# Role of specialized composition of SWI/SNF complexes in prostate cancer lineage plasticity

Joanna Cyrta et al.[#]

Advanced prostate cancer initially responds to hormonal treatment, but ultimately becomes resistant and requires more potent therapies. One mechanism of resistance observed in around 10–20% of these patients is lineage plasticity, which manifests in a partial or complete small cell or neuroendocrine prostate cancer (NEPC) phenotype. Here, we investigate the role of the mammalian SWI/SNF (mSWI/SNF) chromatin remodeling complex in NEPC. Using large patient datasets, patient-derived organoids and cancer cell lines, we identify mSWI/SNF subunits that are deregulated in NEPC and demonstrate that SMARCA4 (BRG1) overexpression is associated with aggressive disease. We also show that SWI/SNF complexes interact with different lineage-specific factors in NEPC compared to prostate adenocarcinoma. These data point to a role for mSWI/SNF complexes in therapy-related lineage plasticity, which may also be relevant for other solid tumors.

[#]A list of authors and their affiliations appears at the end of the paper.

Prostate cancer (PCa) is the second most commonly diagnosed cancer and the fifth cause of cancer-related death in men worldwide[1,2]. Although most men are effectively treated by local therapies (surgery and/or radiotherapy), some develop metastatic recurrence or present with metastases at initial diagnosis. The mainstay of treatment for metastatic PCa is androgen deprivation therapy (ADT), but resistance ultimately develops with progression to castration-resistant prostate cancer (CRPC), which typically harbors a "luminal" (adenocarcinoma) phenotype (CRPC-Adeno) with continued dependence on androgen receptor (AR) signaling[3–5]. Improved, more potent androgen receptor signaling inhibitors (ARSi) have been developed to treat patients that are not responsive to these therapeutics, yet acquired resistance to these drugs ultimately develops as well. In CRPC, indifference to AR signaling may manifest with a distinct histomorphology and expression of neural-like markers, leading to neuroendocrine or small cell prostate cancer (CRPC-NE)[5–7]. Approximately 10–20% of CRPC cases treated with ARSi display a neuroendocrine phenotype[5,8,9]. CRPC-NE no longer responds to ARSi and carries a dismal prognosis, with a mean overall survival of 12 months and no specific standard of care treatment options available[10]. There is mounting evidence that CRPC-Adeno can transdifferentiate to an AR-indifferent state through a mechanism of lineage plasticity under specific genomic conditions, including but not limited to TP53, RB1, and PTEN loss[4,11–13]. Epigenetic regulators, such as EZH2, are also critical in this process[4,12,13]. Although the mammalian Switch Sucrose Non-Fermenting (mSWI/SNF) complex is another major chromatin regulator well known for its role in physiological processes and frequently altered in cancer[14–16], its putative implication in NEPC lineage plasticity is unknown.

Mammalian SWI/SNF complexes, also known as Brg/Brahma-associated factor (BAF) complexes, are a heterogeneous family of ATP-dependent chromatin remodeling complexes composed of about 11–15 protein subunits and generally considered as positive mediators of chromatin accessibility[16]. These complexes are evolutionarily conserved in eukaryotes and required for normal embryonic development[16,17]. Specialized complex assemblies with distinct functions have been identified at different stages of embryogenesis and during tissue maturation[18–22]. Over 20% of human malignancies carry a genomic alteration involving at least one of the SWI/SNF subunit genes[14–16], including malignant rhabdoid tumors[23], synovial sarcoma[24], small cell carcinoma of the ovary hypercalcemic type, ovarian clear cell carcinoma, endometrioid carcinoma, bladder cancer, renal cell carcinoma, and lung adenocarcinoma, among others[14,23,25–27].

To date, SWI/SNF alterations have not been studied in the context of advanced PCa. In this study, we show that SWI/SNF composition is altered in the setting of CRPC-NE and that in contrast to many of the above-cited tumor types, SWI/SNF can have tumor-promoting functions in PCa. We also provide evidence that SWI/SNF interacts with different lineage-specific partners throughout PCa transdifferentiation. Collectively, these findings suggest that specialized SWI/SNF complexes are associated with PCa disease progression and may play a role in therapy resistance.

## Results

**SWI/SNF subunit expression is altered in CRPC-NE.** To define somatic mutation frequencies of genes encoding SWI/SNF subunits across the entire spectrum of PCa, we conducted a comprehensive analysis of whole exome sequencing (WES) data from 600 PCa patients representing a wide range of the disease spectrum, including 56 CRPC-NE cases (Fig. 1a, Supplementary Data 1, Supplementary Data 2, Supplementary Data 3). No

recurrent SWI/SNF somatic mutations were observed and there was a low overall rate of point mutations and insertions/deletions in those genes (59 samples, 9.8% of all cases) (Fig. 1b). We observed an increased percentage of loss-of-heterozygosity (LOH) by hemizygous deletion or copy number neutral LOH (CNNL), in 27 out of 28 genes (significant for 15 genes, proportion test, alpha = 0.05), when comparing localized hormone treatment-naïve PCa vs. CRPC-Adeno (Supplementary Fig. 1, Supplementary Data 1). A similar result was obtained when comparing localized hormone treatment-naïve PCa and CRPC-NE cases (26 out of 28 genes with higher LOH frequency in CRPC-NE). Conversely, there were fewer differences when comparing CRPC-Adeno and CRPC-NE. A significant increase in the fraction of LOH in CRPC-NE as compared to CRPC-Adeno (proportion test, alpha = 0.05) was only noted for three genes: BRD7 (51% vs. 30%, respectively, p = 0.005), SMARCD1 (11% vs. 3%, p = 0.04), and PBRM1 (18% vs. 8%, p = 0.049) (Fig. 1b). However, this was not accompanied by a decrease in SMARCD1 or PBRM1 expression in CRPC-NE (Supplementary Fig. 2). Expression levels of BRD7 were significantly lower in CRPC-NE compared to CRPC-Adeno, but not in CRPC-NE compared to localized PCa. This is in line with a previous study in which BRD7 loss was identified as part of a larger heterozygous deletion event enriched in CRPC-NE and centered around the CYLD gene[4]. Collectively, these observations suggest that the increased fractions of LOH observed in CRPC-NE for BRD7, PBRM1, and SMARCD1 are unlikely to carry functional significance.

Given the modest differential abundance of genomic lesions, we next queried the expression levels of SWI/SNF subunits by examining RNA-seq data of 572 unique PCa patients, including 20 CRPC-NE cases[4,5] (Supplementary Data 4). The SMARCA4 ATPase subunit was significantly upregulated, with accompanying downregulation of its mutually exclusive paralogue SMARCA2[16,28] in CRPC-NE (n = 20) compared to CRPC-Adeno (n = 120) with a mean difference of 0.55 (p = 0.015) (averaged log2(FPKM + 1)) for SMARCA4 and mean difference of −0.60 (p = 0.02) for SMARCA2, respectively (Fig. 1c). A concordant result was observed when comparing SMARCA4/SMARCA2 expression ratios per patient in CRPC-Adeno (median ratio = 1.07) and in CRPC-NE (median ratio = 3.06, p = 0.007) (Supplementary Fig. 3). To validate that these transcriptomic findings translated into differences in protein expression, we performed immunohistochemistry (IHC) on patient samples and confirmed higher SMARCA4 (BRG1) and lower SMARCA2 (BRM) expression with increasing PCa disease progression, with highest SMARCA4 expression observed in CRPC-NE (Fig. 1d and Supplementary Fig. 4).

Importantly, we also identified strong upregulation of neuron-specific SWI/SNF subunit genes in CRPC-NE: ACTL6B (BAF53B), DPF1 (BAF45B), and SS18L1 (CREST) (mean log2 [FPKM + 1] values: 2.79, 1.19, and 3.58, respectively) compared to CRPC-Adeno (mean 0.24, p = 4.86e−06; mean 0.35, p = 0.0016; and mean 2.76, p = 6.85e−05, respectively) (Fig. 1c). These subunits are expressed in post-mitotic neurons, serving instructive functions in neuronal differentiation[22]. By IHC, BAF53B, and BAF45B were highly expressed in CRPC-NE, but absent from benign prostate, localized PCa or CRPC-Adeno samples (Fig. 1d), demonstrating high specificity for the neuroendocrine phenotype.

We also noted intra-tumor heterogeneity in the expression of SWI/SNF subunits, as illustrated by IHC in patient specimens with a mixed phenotype (combining areas with adenocarcinoma and neuroendocrine differentiation) (Supplementary Figs. 4 and 5) and in 3D CRPC-NE organoid cultures (Supplementary Fig. 6). In the latter, we identified distinct cell clusters with high expression of the neural stem cell factor SOX2, low expression

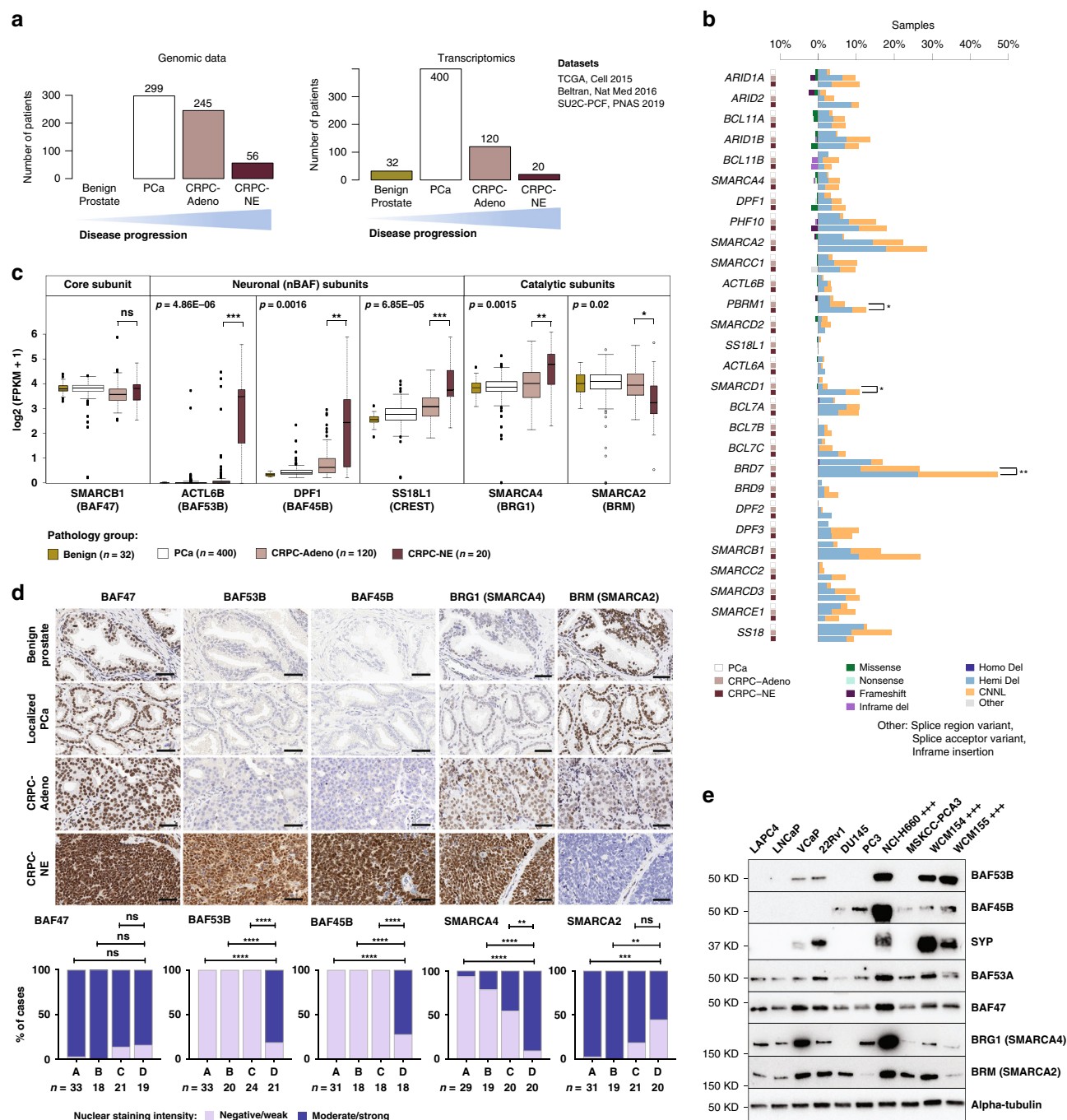

**Fig. 1 Identification of SWI/SNF subunits deregulated in CRPC-NE. a** Summary of the number of patients analyzed by whole exome sequencing (WES) and RNA-seq for each disease state. **b** WES results for SWI/SNF genes in 600 samples from unique PCa patients. For each gene, three consecutive bars represent alteration frequency in localized hormone treatment-naïve PCa, CRPC-Adeno and CRPC-NE, respectively. **c** RNA-seq analysis of gene expression levels in 572 unique patient samples from four studies, showing selected genes (*ACTL6B*: $p = 4.86E-06$, *DPF1*: $p = 0.0016$, *SS18L1*: $p = 6.85E-05$, *SMARCA4*: $p = 0.0015$, *SMARCA2*: $p = 0.02$) significantly deregulated in CRPC-NE. The core subunit *SMARCB1* is shown for comparison. The box plots represent the median values and the lower and upper interquartile range (IQR); the upper whisker = min(max(x), Q3 + 1.5 × IQR) and lower whisker = max (min(x), Q1 − 1.5 × IQR), and the outliers are plotted as individual points. **d** Representative immunostainings against BAF47 (*SMARCB1*), BAF53B (*ACTL6B*), BAF45B (*DPF1*), BRG1 (*SMARCA4*) and BRM (*SMARCA2*), and statistical analysis of staining intensity in patient samples. A-benign prostate glands, B-hormone treatment-naïve localized PCa, C-CRPC-Adeno, D-CRPC-NE. **p < 0.01 ($p = 0.0057$ for BRG1, $p = 0.0012$ for BRM), ***p < 0.001 ($p = 0.0004$ for BRM) and ****p < 0.0001, ns indicates not significant (two-sided Fisher's exact test). Scale bars, 50 μm. **e** Immunoblot showing expression levels of selected SWI/SNF subunits in PCa cell lines (+++ designates CRPC-NE cell lines). Benign: benign prostatic tissue, PCa: localized hormone treatment-naïve prostate cancer, CRPC-Adeno: Castration resistant prostate cancer, adenocarcinoma subtype, CRPC-NE: Castration resistant prostate cancer, neuroendocrine subtype. Source data are provided in the Source Data file.

of the terminal neuronal marker synaptophysin, and higher expression of SMARCA4 (BRG1) and SMARCC1 (BAF155) than in the rest of the cell population (Supplementary Fig. 6). Overall, these observations suggest a relationship between expression of specific SWI/SNF subunits and different phenotype states, which can be seen even in a clonal tumor population.

BAF53B and BAF45B protein expression was confirmed in CRPC-NE cell lines and organoids (NCI-H660, WCM154, and WCM155[29]) (Fig. 1e). BAF53B was also detected, albeit at lower levels, in two synaptophysin-positive PCa cell lines VCaP and 22Rv1, which bear some degree of transcriptomic similarity to neuroendocrine PCa cell lines[9]. BAF45B, on the other hand, was detected in some CRPC-Adeno cell lines and organoids (DU145, PC3, and MSKCC-PCA3). Unlike what we observed in patient samples, we did not observe high SMARCA4 (BRG1) and low SMARCA2 (BRM) expression in CRPC-NE cell lines, which could in part be due to marked differences in cell growth rates among different cell lines (Fig. 1e).

Although in neurons, BAF53B has been characterized as a mutually exclusive paralog to BAF53A, our data revealed that in CRPC-NE, BAF53A expression is maintained (Fig. 1e, Supplementary Fig. 7). BAF53B expression in neurons is known to be mediated by the downregulation of the RE1-Silencing Transcription factor (REST), a negative regulator of neuron-specific genes[20]. In prostate adenocarcinoma cells, we observed that short-term REST knock-down led to an increase of BAF53B (ACTL6B) mRNA and protein levels, but the effect was modest, while other neuronal genes known to be negatively controlled by REST (e.g., synaptophysin) were highly upregulated (Supplementary Fig. 8).

To understand whether high SMARCA4 expression in CRPC-NE was related to other characteristics of CRPC-NE, such as acquisition of pluripotent stem cell-like features, and not only to the expression of terminal neural markers, we performed single-cell RNA-seq on two CRPC-NE organoids in 3D culture (MSKCC PCa1 and 16) and confirmed that SMARCA4 expression was significantly higher in cells with high expression of the pluripotent stem cell marker SOX2 (Supplementary Fig. 9), consistent with our IHC findings (Supplementary Fig. 6). Bulk RNA-seq data from 18 PCa organoids (CRPC-Adeno and CRPC-NE) (Supplementary Fig. 10) revealed that SMARCA4 expression was positively correlated with the expression of synaptophysin (a terminal neuronal marker), but also showed a tendency towards positive correlation with SOX2; conversely, there was a trend towards an inverse correlation between SMARCA2 and SOX2 (Pearson correlation analysis). Of note, some organoids (including MSKCC PCa1 and PCa16) classified as CRPC-NE based on their transcriptomic NEPC score[4] showed high expression of SOX2, but low expression of terminal neural markers, such as synaptophysin (SYP) (Supplementary Fig. 10). These results suggest that high SMARCA4 expression may be related to pluripotent stem cell-like features and/or to proliferation at least in some CRPC-NE, rather than just to the expression of terminal neuronal markers.

Taken together, the above observations suggest that specialized SWI/SNF composition varies with PCa lineage plasticity to small cell or neuroendocrine states.

**High SMARCA4 (BRG1) expression is associated with aggressive PCa.** We posited that high SMARCA4 expression is associated with a more aggressive clinical course. To address this, we interrogated protein expression of SMARCA4 (BRG1) by IHC in a cohort of 203 men operated for localized hormone-treatment naïve PCa (demographics previously described in Spahn et al.[30]). High SMARCA4 protein expression in primary PCa was associated with a significantly shorter overall survival (HR = 2.17 [95% CI: 1.07–4.42], $p = 0.028$) (Fig. 2a). This relationship remained significant after adjustment for single covariates that have known association with PCa outcome (Supplementary Table 1). Patients with high tumor SMARCA2 (BRM) protein expression showed a trend towards a better overall survival, although this relationship did not reach statistical significance. Taken together, the above findings suggest that high SMARCA4 expression is associated with more aggressive cases of PCa.

We next sought to determine the effects of SMARCA4 and SMARCA2 depletion in PCa cell lines. We performed siRNA-mediated knock-down of SMARCA4 and SMARCA2 in an androgen-sensitive (LNCaP) cell line and in a CRPC-Adeno cell line (22Rv1) and compared global transcriptional alterations using RNA-seq. As expected, given its posited dominant role, SMARCA4 depletion demonstrated a stronger effect on the transcriptome of both cell lines, while SMARCA2 depletion led to only modest transcriptional alterations (Fig. 2b, Supplementary Figs. 11 and 12). Among the genes most significantly deregulated upon SMARCA4 knock-down were several of known significance in PCa progression, including: upregulation of cell cycle regulators CDKN1A (p21) and BTG2 (in both LNCaP and 22Rv1 cell lines), downregulation of E2F targets (in both cell lines), downregulation of EZH2, and downregulation of the oncogenic long non-coding RNA PCAT-1 (both significant in LNCaP only)[4,31,32] (Fig. 2c–e, Supplementary Fig. 12, Supplementary Data 5, Supplementary Data 6). We also observed a significant enrichment in gene sets related to EZH2 knock-down, suggesting that knock-down of SMARCA4 and knock-down of EZH2 can have partly overlapping effects in PCa cells (Supplementary Fig. 13). Expression of REST was not altered by SMARCA4 knock-down (Supplementary Fig. 14).

The observed changes in cell cycle-related pathways led us to explore the requirement for SMARCA4 and SMARCA2 for PCa cell growth. Depletion of SMARCA4, but not of SMARCA2, significantly reduced proliferation of the adenocarcinoma cell line LNCaP and the LNCaP-derived androgen-independent CRPC-Adeno cell line C4-2 (Fig. 2f), in line with previous findings[33,34]. Knock-down of SMARCA4, but not of SMARCA2, in PCa cells resulted in a decrease of other SWI/SNF subunits, including SMARCC1 (BAF155) and ACTL6A (BAF53A), at the protein level, but not at the transcript level (Supplementary Fig. 15). Accordingly, both LNCaP and C4-2 cells proved to be highly sensitive to depletion of BAF155 (SMARCC1) (Supplementary Fig. 16). Recent work has shown that sensitivity of PCa cells to SMARCA4 knock-down may be dependent on PTEN loss, via a mechanism of synthetic lethality[33]. To expand upon these findings, we performed knock-down of BAF155 (SMARCC1) in two PTEN wild-type cell lines, 22Rv1 (CRPC-Adeno) and WCM154 (CRPC-NE), and observed a significant decrease in cell growth (Supplementary Fig. 17). This suggests that PTEN-competent PCa cells can still be sensitive to SWI/SNF disruption, even though they may be differentially responsive to depletion of different subunits.

Given that loss of TP53 and/or RB1 has been suggested to confer a poised pluripotent state required for neuroendocrine transdifferentiation[11,12], we also tested the effect of SMARCA4 knock-down in LNCaP cells having undergone CRISPR-Cas9 mediated knock-out of TP53, RB1, or both genes. The effect of SMARCA4 knock-down on cell proliferation was not entirely abrogated by the absence of functional p53 and/or Rb (Supplementary Fig. 18).

To strengthen the above observations of a putative tumor-promoting function of SMARCA4 (BRG1) in PCa, we also sought to study the effects of SMARCA4 overexpression in PCa cells.

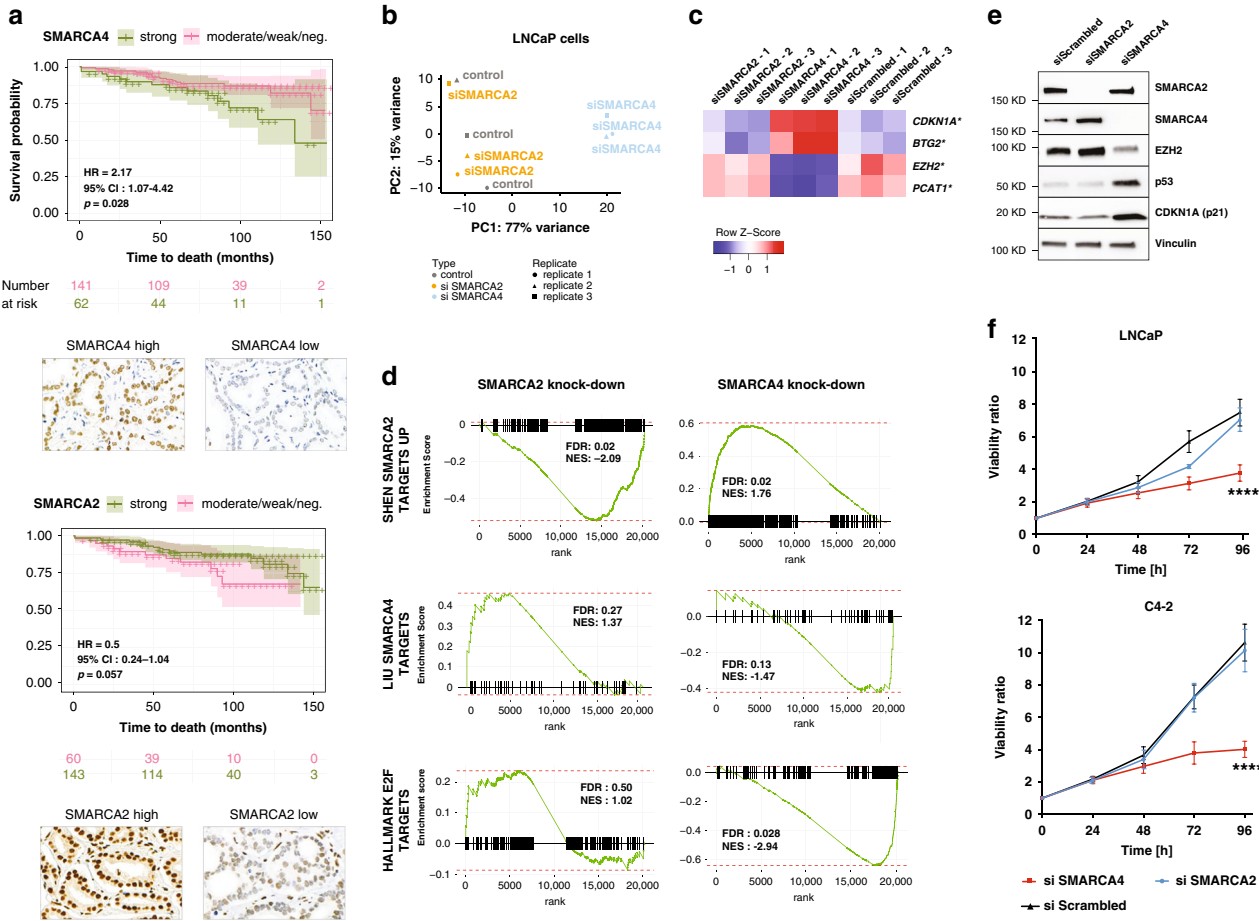

**Fig. 2 SWI/SNF *SMARCA4* and *SMARCA2* expression in prostate cancer. a** Kaplan–Meier curves showing the association between overall survival and SMARCA4 (BRG1) IHC expression ($p = 0.028$, Log-rank test) or SMARCA2 (BRM) IHC expression (not significant), in 203 patients with localized PCa. **b** Principal component analysis (PCA) of RNA-seq data for prostate adenocarcinoma (LNCaP) cells 72 h after *SMARCA4* or *SMARCA2* knock-down. **c** Expression levels (RNA-seq) of selected genes upon *SMARCA4* and *SMARCA2* knock-down in LNCaP cells; *FDR < 0.05. **d** Gene Set Enrichment Analysis based on RNA-seq gene expression analysis in LNCaP cells with *SMARCA4* or *SMARCA2* knock-down. **e** Immunoblot showing selected deregulated proteins upon *SMARCA4* and *SMARCA2* knock-down in LNCaP cells. **f** Effect of *SMARCA4* or *SMARCA2* knock-down on cell proliferation of prostatic adenocarcinoma (LNCaP) and CRPC-Adeno (C4-2) cells. $N = 3$ independent experiments. Data are presented as mean values +/− SEM and analyzed using two-way Anova (****$p < 0.0001$). Statistical significance was evaluated at 0.05 alpha level with GraphPadPrism, version 8.2.1, Mac. Source data are provided in the Source Data file.

22Rv1 cells were stably transduced with lentiviral vectors designed to overexpress either *SMARCA4* or *SMARCA2* or with a matched empty control vector, and sorted based on the expression of the fluorescent reporter. Despite strong expression of the reporters, we did not observe an increase in SMARCA4 or SMARCA2 at the protein level (Supplementary Fig. 19). However, after an additional 24 h treatment with the proteasome inhibitor MG-132, SMARCA4, and SMARCA2 overexpression was readily detected. These findings hint towards a tight and context-dependent regulation of catalytic SWI/SNF subunits, as forced isolated overexpression of a single subunit seems to provoke rapid degradation of the excess protein. Thus, it is possible that SMARCA4 overexpression may be necessary, but not sufficient, to promote an aggressive phenotype in prostate cancer cells.

To understand whether BAF53B and BAF45B—two other subunits overexpressed in CRPC-NE—potentially regulated similar gene expression programs as *SMARCA4*, we performed shRNA-mediated knock-down of these subunits in the CRPC-NE organoid line WCM155. Neither BAF53B nor BAF45B knock-down had an effect on CRPC-NE cell proliferation (Supplementary Fig. 20) Therefore, it appears that BAF53B and BAF45B expression may be specific for the CRPC-NE phenotype, but not a

critical mediator of CRPC-NE aggressiveness. Collectively, the above genomic, transcriptomic, and functional findings support a tumor-promoting role of *SMARCA4*-containing mSWI/SNF complexes in PCa.

**Aggressive prostate cancer anti-correlates with SMARCA4 knock-down signature**. Based on the association of *SMARCA4* expression with poor clinical outcome, and observations that *SMARCA4* knock-down leads to a significant decrease in PCa cell growth in line with previous studies[33,34], we posited that a *SMARCA4* knock-down signature (composed of genes deregulated upon *SMARCA4* depletion) would be associated with more indolent PCa. To address this, we interrogated RNA-seq data of several large clinical cohorts using a *SMARCA4* knock-down signature derived from the LNCaP PCa cell line (see "Methods" section) and composed of the top 419 deregulated genes. A high *SMARCA4* knock-down signature score was, indeed, associated with more indolent disease. In contrast, a low *SMARCA4* knock-down signature score was associated with more aggressive PCa.

As expected, a low *SMARCA4* knock-down signature score was also strongly associated with a CRPC-NE phenotype. We

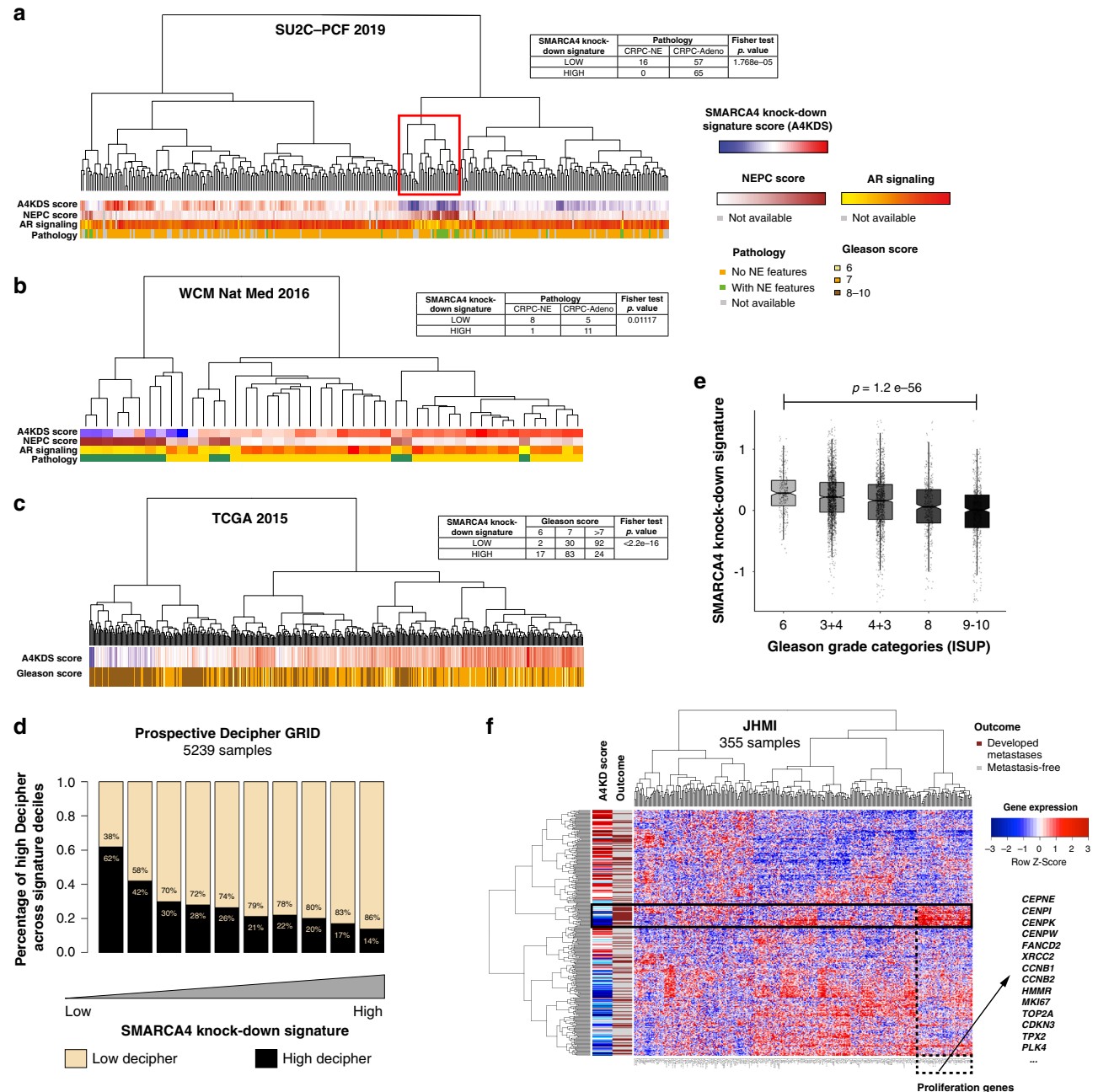

**Fig. 3 Transcriptomic *SMARCA4* knock-down signature in PCa cohorts. a** 332 cases of CRPC from the SU2C-PCF cohort (table: *n* = 138). **b** 47 cases of CRPC from the WCM cohort (table: *n* = 25). **c** 495 cases of localized PCa from the TCGA cohort (table: *n* = 248); *p* value = 1.21e−18 (**a–c** represent two-paired tests). **d** Low *SMARCA4* knock-down signature scores are associated with high Decipher scores (surrogate for risk of metastasis) in 5239 primary PCa samples from the Prospective Decipher GRID (Mann–Whitney *U* test). **e** Low *SMARCA4* knock-down signature scores are associated with higher Gleason score in the same Decipher GRID cohort (Mann–Kendall trend test). The center of each boxplot represents the mean, lower bound represents the 25th percentile, the top bound represents the 75th percentile, the whiskers represent the 95% CI. *p* value = 1.2e−56. **f** Unsupervised clustering of patients from in the JHMI natural history PCa cohort (Johns Hopkins Medical Institute, *n* = 355) based on the downregulated genes from the *SMARCA4* knock-down signature, and compared to metastatic outcome (brown: metastatic recurrence, gray: metastasis-free). Overexpression of a subset of genes, many of which are related to proliferation, is seen in a cluster of patients who presented metastatic outcome (black box).

examined two CRPC cohorts consisting of 332 patients from the Stand Up To Cancer-Prostate Cancer Foundation (SU2C-PCF) trial treated with ARSi[5] and 47 patients from the Weill Cornell Medicine (WCM) cohort[4]. In the SU2C-PCF cohort, when considering patients from the highest (top 25%) and lowest (bottom 25%) quartiles of *SMARCA4* knock-down signature scores (*n* = 138), low *SMARCA4* knock-down signature scores were significantly more often observed in CRPC-NE cases (*n* = 16

or 100%) than in CRPC-Adeno cases (*n* = 57 or 46.7%) (*p* = 1.77e−05) (Fig. 3a). A similar result was obtained in the WCM cohort (*n* = 25): low *SMARCA4* knock-down signature scores were seen in 89% (*n* = 8) of CRPC-NE cases vs. 31% (*n* = 5) of CRPC-Adeno cases (*p* = 0.011) (Fig. 3b). Furthermore, low *SMARCA4* knock-down signature was associated with a higher NEPC[4] and a lower AR signaling score[35] in both cohorts (Supplementary Table 2). One particularly informative cluster

was found to show low *SMARCA4* knock-down signature scores, high CRPC-NE scores, and low AR signaling scores (Fig. 3a, red box). Of note, *SMARCA4* mRNA levels were consistent with the predicted signature score in all analyzed cohorts (Supplementary Fig. 21).

We next queried if the *SMARCA4* knock-down signature was associated with higher tumor grade, referred to as Gleason score risk groups in localized PCa[36]. We first explored 248 patients from The Cancer Genome Atlas (TCGA) PCa cohort with localized, hormone treatment-naïve PCa[37]. Tumors in the highest Gleason score risk groups (IV and V) more often displayed low *SMARCA4* knock-down signature scores ($p < 2.2e-16$) (Fig. 3c).

As high tumor grade is associated with risk of metastatic progression, we decided to validate these findings in other independent clinical cohorts annotated with clinical survival data. We calculated *SMARCA4* knock-down signature scores for 5239 prospectively collected radical prostatectomy samples from men with localized PCa and analyzed with the Decipher GRID transcriptomic platform[38]. Samples with a low *SMARCA4* knock-down signature (lowest 10%) were significantly enriched (62%) with high Decipher score, which is a strong surrogate of metastasis prediction[38] (Fig. 3d), compared to 14% in samples with high *SMARCA4* knock-down signature (highest 10%). In this patient population and consistent with TCGA results, we observed an association between *SMARCA4* knock-down signature and Gleason score risk categories: signature scores in the Gleason 9–10 group (mean = −0.13) were significantly lower compared to the Gleason 6 group (mean = 0.29, $p = 1.2$ e−56) (Fig. 3e). We next explored an independent retrospective cohort from Johns Hopkins Medical Institution (JHMI)[39]. In the JHMI cohort, patients with low *SMARCA4* knock-down signature showed a trend towards higher metastasis frequency, the strongest surrogate for lethal disease progression (Supplementary Fig. 22). When clustering patients based on the downregulated genes (Fig. 3f) or on all genes (Supplementary Fig. 23) that make up the *SMARCA4* knock-down signature, overexpression of a subset of genes involved in cell proliferation was associated with a cluster of patients enriched with metastatic outcome (Fig. 3f, box). In summary, these results from large patient cohorts confirm that the lowest *SMARCA4* knock-down signatures are observed in the most aggressive PCa.

**The SWI/SNF complex has distinct lineage-specific interaction partners in CRPC-NE and in prostate adenocarcinoma cells.** To gain insight into the potential effectors of NEPC-specific epigenetic regulation, we next sought to identify interactors of mSWI/SNF in the context of CRPC-NE and prostate adenocarcinoma cell lines. To this end, we performed co-IP with an antibody directed against the core SWI/SNF subunit BAF155 (*SMARCC1*) at low stringency (see "Methods" section) followed by mass spectrometry (MS) in NCI-H660 (a CRPC-NE cell line) and in LNCaP-AR cells (LNCaP cells engineered to overexpress the androgen receptor[40]). Proteins that immunoprecipitated with BAF155 in CRPC-NE cells, but not in adenocarcinoma cells, (Fig. 4a, b) included BAF53B (ACTL6B) and BAF45B (DPF1) subunits, as anticipated from results described above, as well as several factors specific to neural differentiation, such as the transcription factor NKX2.1 (TTF-1), the microtubule-associated factor MAP2 and the growth factor VGF. Moreover, we found several members of the NuRD chromatin remodeling complex, such as MTA1 and CHD4, to immunoprecipitate with BAF155. This is in line with previous findings of a potential interaction of those two chromatin remodeling complexes (Fig. 4a, b)[41,42]. A considerable amount of CRPC-NE specific SWI/SNF interactors were proteins involved in chromatin regulation or DNA repair

(Fig. 4a, b, Supplementary Data 7, Supplementary Data 8). Conversely, proteins that immunoprecipitated with BAF155 in adenocarcinoma cells, but not in CRPC-NE, included HOXB13, a homeobox transcription factor involved in AR signaling[43] (Fig. 4b). In line with these findings, genes encoding most of the above factors were differentially expressed between CRPC-NE and adenocarcinoma cell lines and organoids (Fig. 4c, Supplementary Fig. 24). Further, we confirmed unique interaction of factors NKX2.1, CHD4, MTA1, and VGF with BAF155 in NCI-H660 by immunoblotting, while these interactions were absent in LNCaP-AR cells (Supplementary Fig. 25a). Interaction of HOXB13 with BAF155 in LNCaP-AR cells was also confirmed by immunoblotting (Supplementary Fig. 25b). The co-IP experiment also showed an enrichment of proteins negatively associated with REST signaling in NCI-H660 cells, such as HMG20A, a chromatin-associated protein known to overcome the repressive effects of REST and induce activation of neuronal genes[44]. Loss of expression or altered splicing of REST has been associated with neural-like lineage plasticity in PCa in multiple studies[45–51]. An independent co-IP experiment using an antibody directed against SMARCA4 followed by MS in NCI-H660 and in LNCaP cells found similar results for BAF53B, BAF45B, NKX2.1, and HOXB13 (Supplementary Fig. 26, Supplementary Data 9).

As a proof-of-principle, we compared genome occupancy of SMARCC1, HOXB13, the active chromatin histone mark H3K27ac and the inactive chromatin mark H3K27me3 in LNCaP cells, using published ChIP-seq datasets (Supplementary Fig. 27). SMARCC1 and HOXB13 colocalized at active chromatin sites (11,824 sites), while there was almost no overlap between SMARCC1 and HOXB13 at inactive chromatin sites, thus suggesting a functional nature of this interaction. Collectively, the above observations suggest that the set of SWI/SNF interaction partners in CRPC-NE is quite distinct from the one in prostatic adenocarcinoma.

## Discussion

Whereas neuroendocrine PCa is rarely present at diagnosis in hormone-treatment naïve PCa patients (de novo neuroendocrine PCa, <1% of cases)[52], recent work supports the hypothesis that acquisition of a CRPC-NE phenotype in PCa is a more common mechanism of resistance to ARSi[4,5,8,13,53]. Based on a recent review of 440 CRPC patients, CRPC-NE was seen in 11% of CRPC patients that underwent biopsy[5,8,9]. There is increasing evidence that CRPC-NE can directly arise from CRPC-Adeno cells through lineage plasticity (Supplementary Fig. 28), which is supported by lineage tracing experiments in a genetically engineered mouse model of PCa with combined *Trp53* and *Pten* loss[54]. Moreover, mouse models with *Trp53* and *Rb1* genomic loss show lineage plasticity, but epigenetic therapy can re-sensitize those tumors towards ARSi treatment[11]. In patient cohorts, CRPC-NE are characterized by an overexpression of several epigenetic regulators (such as EZH2) and a specific DNA methylation profile[4,13,29]. Overall, these data support the idea that PCa progression through lineage plasticity is regulated by epigenetic changes in a specific genomic context[12,55].

Given that mSWI/SNF complexes are major epigenetic regulators in physiological cell differentiation, we posited that they may play a role in CRPC-NE lineage plasticity. Specialized assemblies of the SWI/SNF complex with distinct functions are observed at different stages of embryonic development and tissue maturation[18,19]. The most notable changes in SWI/SNF composition described to date occur during neuronal differentiation. Cells committed to the neural lineage initially express a neural progenitor form of the complex (termed npBAF), which incorporates among others the BAF53A, BAF45A/D, and

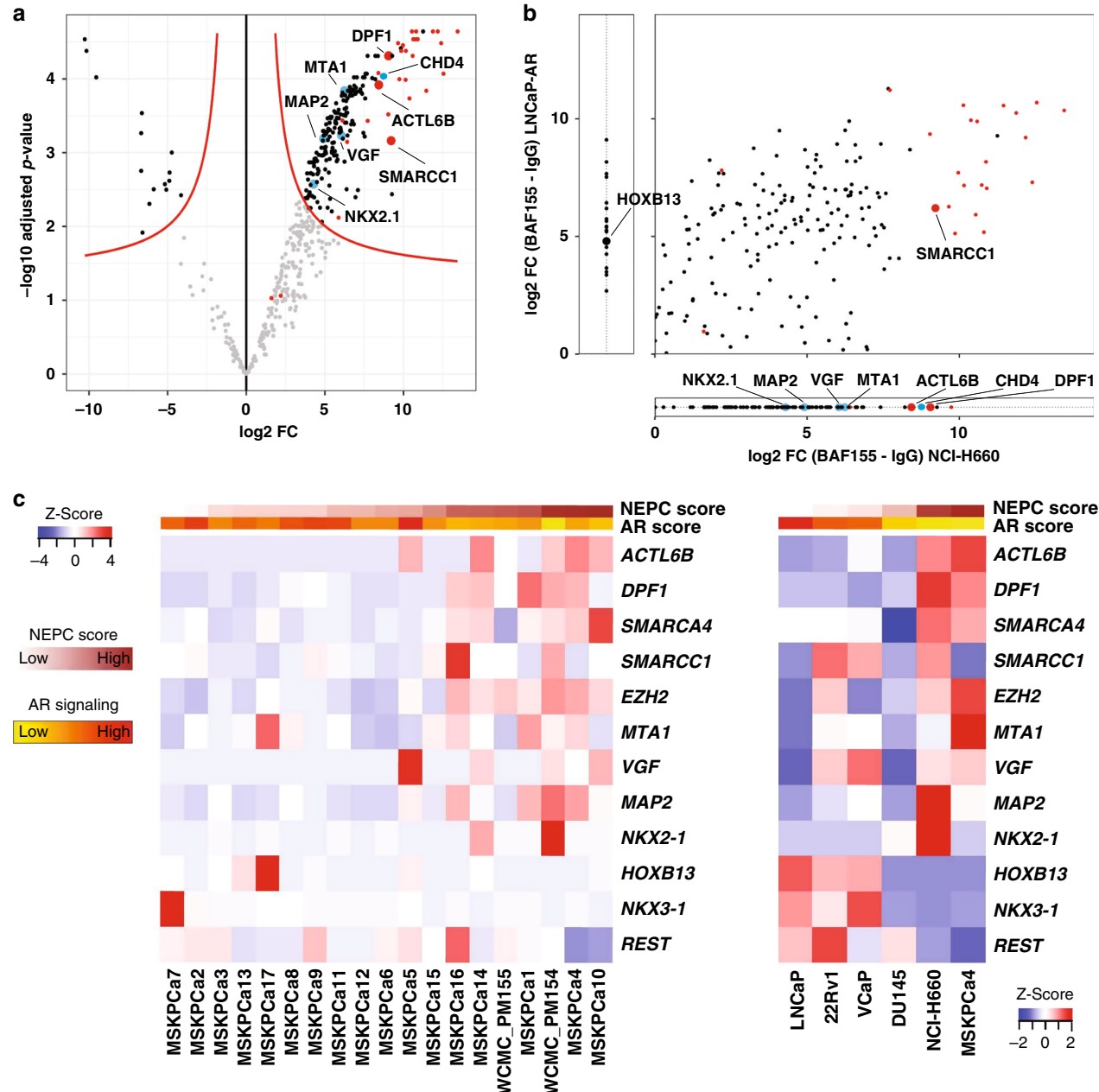

**Fig. 4 SWI/SNF associates with different transcriptional regulators in CRPC-NE and in adenocarcinoma cells. a** Volcano plot showing proteins most significantly represented (upper right) in the co-IP using an anti-BAF155 antibody, as compared to IgG isotype control in NCI-H660 (CRPC-NE) cells (pooled data from 3 co-IP replicates). The x-axis represents log2 fold change (FC) values, the y-axis represents −log10 of adjusted p-values. Each dot represents a protein; red dots represent SWI/SNF members, blue dots indicate notable findings. **b** A qualitative representation comparing proteins associated with SWI/SNF in NCI-H660 (CRPC-NE) and in LNCaP-AR (adenocarcinoma) cells (averaged data from two co-IP experiments). Plotted are log2 fold change values between BAF155 IP and IgG IP in NCI-H660 cells (x-axis) and in LNCaP-AR cells (y-axis), for proteins present in both cell lines with sufficient evidence in each cell line (i.e., if present in two replicates of at least one condition). Proteins plotted outside of the main field represent proteins that were detected exclusively in one of the cell lines. **c** Heatmap showing RNA-seq expression (FPKM) of prostate cancer 3D organoids (left) and 2D cell lines (right), ordered by increasing NEPC score.

SS18 subunits[20–22]. However, upon differentiation to post-mitotic neurons, the complex undergoes a switch to the neural variant and incorporates the respective paralogs of these subunits (i.e., BAF53B, BAF45B/C, and SS18L1). This switch is mediated by repression of BAF53A by micro-RNAs in response to down-regulation of REST[20]. In this study, we observed for the first time the presence of "neuronal" SWI/SNF subunits outside of the nervous system, characterized by the expression of BAF53B and BAF45B in CRPC-NE. Although expression of these subunits was

highly specific of CRPC-NE, it remains unclear whether they play a role in activating neural-like gene programs, or are simply expressed as a consequence of this process. Additional studies are warranted to assess the putative utility of BAF53B and BAF45B as CRPC-NE biomarkers or as predictors of patients at risk of developing CRPC-NE from CRPC-Adeno while on ARSi. Of note, expression of the BAF53A paralogue is retained in CRPC-NE, pointing to potential differences in the way SWI/SNF complexes assemble in post-mitotic neurons and in neuroendocrine

cancer cells, and to possible co-existence of different forms of the complex within the same tumor.

This study supports a pleiotropic role for the SWI/SNF chromatin remodeling complex in cancer, which may depend on the genomic and/or the epigenetic context—a paradigm which has been gaining support both in regards to SWI/SNF and to other epigenetic regulators[56–58]. Although the complex has been described as a tumor suppressor in many cancer types[14,23,25,59], there is increasing evidence for possible tumor-promoting functions of SWI/SNF in other malignancies, including leukemia, breast, liver and pancreas cancer melanoma, glioblastoma, neuroblastoma and synovial sarcoma[24,60–65]. In PCa, the role of SWI/ SNF has long remained insufficiently characterized. Our study provides novel evidence that it can have tumor-promoting functions in PCa, including its most aggressive forms. Based on prior studies and on the current analysis, mutations in SWI/SNF genes are very rare in PCa[4,5,34,37,66–68] (see Fig. 1b), in contrast to some other cancers types[14,15]. From the functional perspective, inhibition of the SWI/SNF subunits BAF57 (SMARCE1) or BAF53A (ACTL6A) in PCa cells has been shown to abrogate androgen-dependent cell proliferation[69,70]. Similarly, Sandoval et al. reported that SWI/SNF interacts with ERG in PCa cells harboring the TMPRSS2:ERG gene fusion and is required to activate specific gene programs to maintain cell growth[71]. Although on the contrary, Prensner et al. had suggested that SWI/ SNF acts as a tumor suppressor in PCa, by demonstrating an antagonistic relationship between the pro-oncogenic long noncoding RNA SChLAP1 and the SWI/SNF core subunit BAF47[72], a subsequent study failed to confirm that SChLAP1-SWI/SNF interaction leads to depletion of SWI/SNF from the genome[73]. Most recently, two studies demonstrated that SMARCA4 was required for growth of prostatic adenocarcinoma cells[33,34], as also confirmed by our results (Fig. 2). Accordingly, localized PCa has been reported to show higher SMARCA4 and lower SMARCA2 expression than benign prostate tissue[33,34,74,75]. We confirm these results and further report an overexpression of SMARCA4 in CRPC and especially in CRPC-NE, in contrast to lower expression in early PCa. In addition, we show that a low SMARCA4 knock-down gene signature score is associated with aggressive PCa, and with a CRPC-NE phenotype.

Recent work by Ding et al. specifically proposed a synthetic lethal association between PTEN and SMARCA4 in PCa, identified through a CRISPR-Cas9 screen[33]. They showed that in vitro, SMARCA4 knock-down leads to decreased cell proliferation in PTEN-negative cell lines, and confirmed these findings in a mouse model. In our study, knock-down of the core SWI/SNF subunit BAF155 (SMARCC1) and BAF170 (SMARCC2) inhibited growth of both PTEN-deficient and PTEN-competent PCa adenocarcinoma cells (Supplementary Figs. 16 and 17), and the PTEN-competent CRPC-NE cell line WCM154 was sensitive to ablation of BAF155, but not of BAF170. This suggests that even if PTEN-competent cells are not sensitive to SMARCA4 loss, they may still be vulnerable to SWI/SNF disruption through depletion of other critical subunits. Taken together, our and previously published findings indicate that PCa expands the spectrum of cancer types in which SWI/SNF can display tumor-promoting functions.

In addition, we observed that SWI/SNF composition in prostate cancer is not a hard-set feature; instead, specialized forms of SWI/SNF may assemble in cancer cells depending on their phenotype (Fig. 5). There is increasing evidence that de-repression of "terminal" neuronal genes in PCa cells is not sufficient to model other critical steps of neuroendocrine lineage plasticity in CRPC-NE[76]. As such, the distinct phenotype of CRPC-NE is not limited to the expression of terminal neuronal markers, but involves other key characteristics, such as dedifferentiation, AR signaling

indifference, acquisition of stem cell-like features and/or high proliferation[13]. In line with this, we show that some patient-derived PCa organoids that are classified as CRPC-NE using a transcriptome-based NEPC score[4] (Supplementary Fig. 10), do not all show high expression of terminal neural markers such as synaptophysin, but instead may highly express factors related to "stemness" (e.g., SOX2). Based on our observations, it is possible that specific forms of SWI/SNF are implicated in various above-mentioned cellular processes, rather than only in the expression of terminal neuronal markers. One possible hypothesis is that an equivalent of the embryonic stem cell form of the complex (esBAF), which is known to exclusively incorporate BRG1 (SMARCA4), BAF53A and BAF155 (SMARCC1) subunits and not their paralogs[18,19], could exist in cancers cells with pluripotent stem cell-like features, and possibly explain the overexpression and/or the functional requirement for these subunits. Similarly, neural-like forms of the complex, including BAF53B and/or BAF45B, could be more specific of cancer cells with a more terminal neural-like phenotype. Further studies are needed to determine whether variants of SWI/SNF can co-exist within the same cell or whether they define distinct tumor sub-populations, in line with what we have observed in 3D CRPC-NE organoid cultures (Supplementary Fig. 6).

One of the ways in which SWI/SNF might contribute to CRPC-NE transdifferentiation is by cooperating with other transcriptional regulators in a context-dependent manner. To this end, we showed that SWI/SNF interacts with different lineage-specific proteins in CRPC-NE than in adenocarcinoma cells (Fig. 4b, Supplementary Fig. 25). In particular, SWI/SNF interacts with the transcription factor NK2 homeobox 1 (NKX2.1/TTF-1) in CRPC-NE cells, but not in adenocarcinoma cells (Fig. 4c, Supplementary Fig. 25). TTF-1 is a master regulator critical for the development of lung and thyroid, but also of specific parts of the brain[77–79] and is known to be expressed in neuroendocrine neoplasms, including CRPC-NE[76]. We also observed SWI/SNF interaction with Metastasis-associated Protein 1 (MTA1), a member of the nucleosome-remodeling and deacetylation complex (NuRD), which is overexpressed in metastatic prostate cancer[80] (Fig. 4c). Conversely, we found HOXB13 to be specifically associated with SWI/SNF in adenocarcinoma cells, and not in CRPC-NE. HOXB13 is a homeobox transcription factor involved in prostate development and displays context-dependent roles in PCa: it can act as a collaborator or a negative regulator of AR signaling[43,81], it cooperates with the AR-V7 splice variant found in a subset of CRPC-Adeno[82], and germline gain-of-function G84E HOXB13 mutations are associated with increased prostate cancer risk[83]. The fact that by ChIP-seq, SWI/SNF colocalizes with HOXB13 at active chromatin sites in prostatic adenocarcinoma cells, further supports the hypothesis that interaction between SWI/SNF and lineage-specific factors in PCa may be meaningful at the functional level.

In conclusion, this work confirms that SWI/SNF has tumor-promoting functions in PCa, including the lethal CRPC-NE. Our findings provide a rationale to further study selected SWI/SNF subunits as potential therapeutic targets in PCa.

## Methods

**Genomic analysis**. Matched tumor and normal WES data of localized and advanced prostate cancer from The Cancer Genome Atlas[84], SU2C-PCF[5] and from the Weill Cornell Medicine cohort[4] were uniformly analyzed for somatic copy number aberrations (SCNA) with CNVkit[85], and for single nucleotide variations (SNVs) and indels with MuTect2[86]. SNVs and Indels were annotated with variant effect predictor (VEP)[87] and only mutations with HIGH or MODERATE predicted impact on a transcript or protein (https://www.ensembl.org/info/genome/variation/prediction/predicted_data.html) were retained. All samples with tumor ploidy and purity estimated using CLONET[88] were retained in the analyses and processed for allele specific characterization. The integrated dataset includes 299

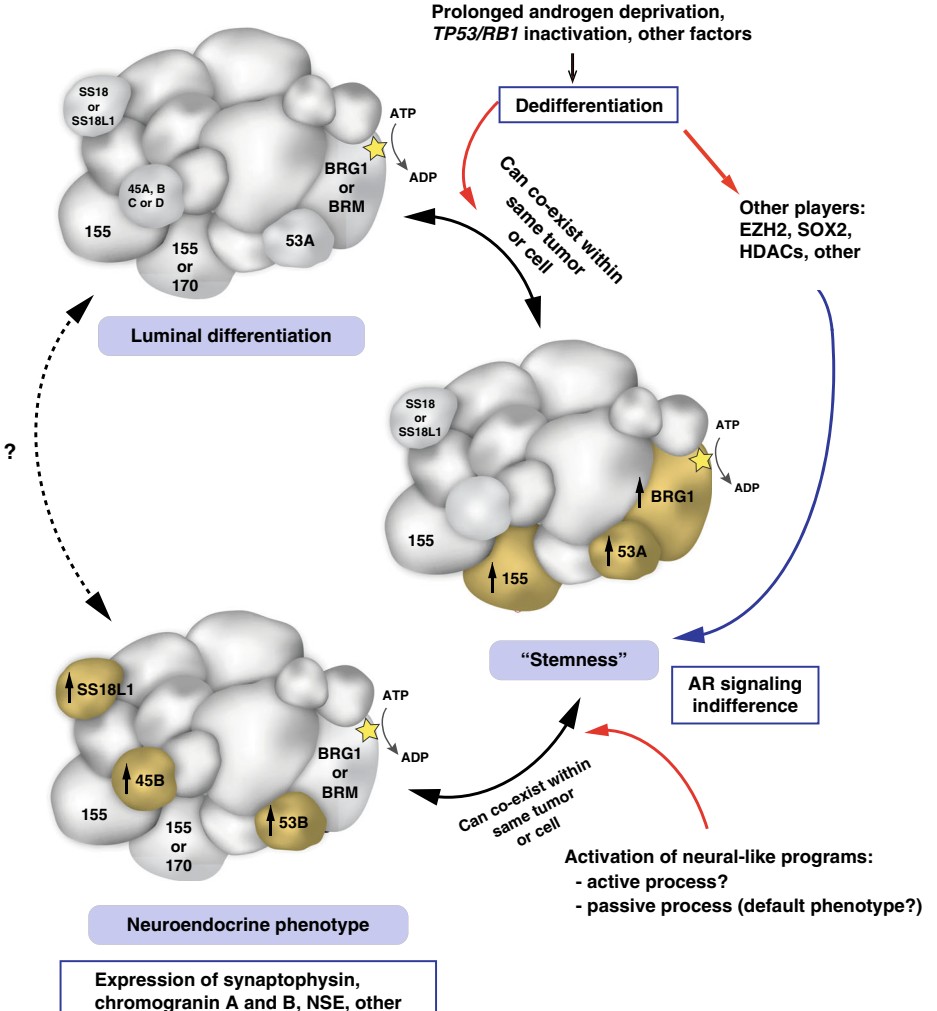

**Fig. 5 Schematic representation of putative specialized SWI/SNF assemblies in prostate cancer cells.** Hypothetical SWI/SNF assemblies are shown in the context of current knowledge about prostate cancer phenotype plasticity. Subunits of particular interest are annotated with their names. Two names within a subunit indicate possible incorporation of either one of the two paralogs. Subunit sizes are approximately indicative of their molecular weights. 155: BAF155, 170: BAF170, 53A: BAF53A, 53B: BAF53B, 45B: BAF45B, AR: Androgen receptor.

unique hormone treatment-naïve prostatic adenocarcinoma (Adeno), 245 castration resistant prostate adenocarcinoma (CRPC-Adeno), and 56 castration resistant neuroendocrine prostate carcinoma (CRPC-NE) patients. Two-tailed proportion test has been used to check enrichment of hemizygous deletion and copy number neutral loss.

**RNA-seq data analysis of human samples**. RNA-seq data from 32 normal prostate samples[89,90], 400 localized PCa[37,89,90] and 120 CRPC-Adenos and 20 CRPC-NE patients[4,5] were utilized for the initial investigation of the SWI-SNF complex units levels and were processed as follows. Reads (FASTQ files) were mapped to the human genome reference sequence (hg19/GRC37) using STAR v2.3.0e[91], and the resulting alignment files were converted into Mapped Read Format (MRF) for gene expression quantification using RSEQtools[92] and GEN-CODE v19 (http://www.gencodegenes.org/releases/19.html) as reference gene annotation set. A composite model of genes based on the union of all exonic regions from all gene transcripts was used, resulting in a set of 20,345 protein-coding genes. Normalized expression levels were estimated as FPKM. After converting the FPKM via log2 (FPKM + 1), differential expression analysis was performed using Mann-Whitney Wilcoxon test. RNA-seq data of the SU2C-PCF cohort were downloaded from original study[5]. NEPC score and AR signaling score were inferred as previously described[5]. Gleason scores of the TCGA PCas were retrieved from the original study[37]. RNA-seq data and Gleason score from the TCGA PCa dataset were retrieved from the TCGA data portal using TCGAbiolinks R package v2.12.2[93].

**Immunohistochemistry**. Immunohistochemistry (IHC) was performed on sections of formalin-fixed paraffin-embedded patient tissue (FFPE) using a Bond III automated immunostainer and the Bond Polymer Refine detection system (Leica Microsystems, IL, USA). Slides were de-paraffinized and heat-mediated antigen retrieval using the Bond Epitope Retrieval 1 solution at pH6 (H1) or Bond Epitope Retrieval 2 solution at pH9 (H2) or enzyme-mediated antigen retrieval (E1) was performed. All antibodies, dilutions and conditions used are listed in Supplementary Table 3.

The intensity of nuclear immunostaining for SWI/SNF subunits was evaluated on tissue micro-arrays (TMAs) and whole slide sections by a pathologist (J.C.) blinded to additional pathological and clinical data, and was scored as negative (score 0), weak (score 1), moderate (score 2), or strong (score 3). Association between disease state and staining intensity (negative/weak vs. moderate/strong) was examined using the two-tailed Fisher's exact test.

**Analysis of *SMARCA4* and *SMARCA2* expression in localized PCa vs. clinical outcome**. The patient cohort with localized PCa and available clinical and follow-up information has been previously described[30]. IHC for SMARCA4 and SMARCA2 was performed on TMAs constructed from these patients' prostatectomy specimens. Staining intensity was scored by a pathologist (J.C.) blinded to the clinical data, using the digital online TMA scoring tool Scorenado (University of Bern, Switzerland). The Kaplan-Meier method was used to estimate patients' overall survival. The association between SMARCA4 and SMARCA2 expression (strong vs. moderate/weak/negative) and overall survival was examined using the log-rank test and multivariable Cox proportional hazards regression models. Ninety-five percent confidence intervals were calculated to assess the precision of the obtained hazard ratios. All *p*-values were two-sided, and statistical significance was evaluated at the 0.05 alpha level. All analyses were performed in R (3.5.1) for Windows.

**Development of a *SMARCA4* knock-down signature**. We defined the *SMARCA4* knock-down signature by selecting a list of differentially expressed genes between *SMARCA4* siRNA-mediated knock-down and Scrambled control in the LNCaP cell line with a log fold change of 1.5 and an FDR < 0.01. For each sample, gene expression data were first normalized by *z*-score transformation. Then signature score was calculated as a weighted sum of normalized expression of the genes in the signature and was finally re-scaled with the 2.5% and 97.5% quantiles equaled −1 and +1, respectively. We defined samples with low *SMARCA4* knock-down signature score as the 25% of cases with the lowest scores, and samples with high signature score as the 25% of cases with the highest scores.

**Validation of *SMARCA4* knock-down signature in multiple clinical cohorts**. *SMARCA4* knock-down generated signature was applied to two CRPC cohorts consisting of 332 patients from the Stand Up To Cancer-Prostate Cancer Foundation (SU2C-PCF) trial treated with ARSi (recently published by Abida et al.[5]) and 47 patients from the Weil Cornell Medicine (WCM) cohort (published by Beltran et al.[4]) and on one cohort of localized, hormone treatment-naïve PCa consisting of 495 patients from The Cancer Genome Atlas (TCGA).

Results from the signature was then correlated with NEPC score and AR signaling scores for the SU2C-PCF and the WCM dataset and with Gleason score for the TCGA dataset.

**Decipher GRID analysis**. For prospective Decipher GRID and JHMI cohort, tumor RNA was extracted from FFPE blocks or slides after macrodissection guided by a histologic review of the tumor lesion by a GU pathologist. RNA extraction and microarray hybridization were all done in a Clinical Laboratory Improvement Amendments (CLIA)-certified laboratory facility (GenomeDx Biosciences, San Diego, CA, USA). Total RNA was amplified and hybridized to Human Exon 1.0 ST GeneChips (Thermo-Fisher, Carlsbad, CA). All data was normalized using the Single Channel Array Normalization (SCAN) algorithm[94]. Decipher scores were calculated based on the predefined 22-markers[38]. Patients with high Decipher (>0.7) were categorized as genomically high risk patients. Mann–Whitney U test was used to assess score differences across Gleason score groups and Mann–Kendall trend test was used to test the association between the percentage of high Decipher scores across deciles of the *SMARCA4* knock-down signature. Kaplan-Meier analysis and Cox proportional hazard model was used to associate *SMARCA4* knock-down signature with time to metastasis in the JHMI cohort.

**Cell culture**. Commercially available PCa cell lines (RWPE-1, LNCaP, 22Rv1, VCaP, LAPC4, PC3, DU145, NCI-H660, C4-2) were purchased from ATCC and maintained according to ATCC protocols. WCM154 and WCM155 CRPC-NE cell lines have been previously established and were maintained in two-dimensional monolayer culture according to the previously described protocol[29]. LNCaP-AR cells were a kind gift from Dr. Sawyers and Dr. Mu (Memorial Sloan Kettering Cancer Center) and were cultured as previously described[12]. MSKCC-PCa3 CRPC-Adeno cells were a kind gift from Dr. Chen (Memorial Sloan Kettering Cancer Center) and were maintained identically to WCM154 and WCM155 cells. All cell lines used and their phenotype are listed in Supplementary Table 4. Cell cultures were regularly tested for *Mycoplasma* contamination and confirmed to be negative.

**Cell transfection and siRNA-mediated knock-down**. ON-TARGET plus siRNA SMARTpool siRNAs against *SMARCA4, SMARCA2, SMARCC1, SMARCC2,* and REST were purchased from Dharmacon. Transfection was performed overnight on attached cells growing in 6-well plates using the Lipofectamine 3000 reagent (Thermo Fisher Scientific) to the proportions of 10 μL of 20 μM siRNA per well. Cells were harvested for protein and RNA extraction 72 h after transfection.

**Cell infection, shRNA-mediated knock-down and gene overexpression**. The *ACTL6B* shRNA and the matching Scrambled shRNA control were a kind gift from Dr. Cigall Kadoch (Dana Farber Cancer Institute). The vector was pGIPZ and the target sequence was: sh#1–TGGATCACACCTACAGCAA. The *DPF1* shRNA and the corresponding Scrambled shRNA control were purchased from Genecopoeia. The vector was psi-LVRU6GP and the target sequences were: sh#1–GAATTAACT TGTTCTGTGTAT, Scrambled control–GCTTCGCGCCGTAGTCTTA. For infection, WCM155 cells were collected, resuspended in media containing Polybrene (Millipore) and lentiviral particles, and centrifuged at 800 × *g* at room temperature for 60 min. Both vectors included a GFP reporter and infection efficiency was confirmed by green fluorescence. Cells were harvested for protein and RNA extraction 72 h after transfection. Given the short-term nature of the experiments, selection was not performed. For the *SMARCA4* or *SMARCA2* overexpression experiment, lentiviral particles were prepared as described above using the pEZ-Lv203 vector (*SMARCA4* gene, eGFP reporter), the pEZLv216 vector (*SMARCA2* gene, mCherry reporter) (all vectors Genecopoeia, MD, USA; all sequence-verified). 22Rv1 cells were infected as described above, cultured and sorted based on the expression of the fluorescent reporter.

**Immunoblotting**. Cells were lysed in RIPA buffer with protease and phosphatase inhibitors (Thermo Fisher Scientific) and total protein concentration was measured

using the DC Protein Assay (Bio-Rad). Protein samples were resolved in SDS-PAGE, transferred onto a nitrocellulose membrane using the iBlot 2 dry blotting system (Thermo Fisher Scientific) and incubated overnight at 4 °C with primary antibodies dissolved in 5% Blotting-Grade Blocker (Bio-Rad). All primary antibodies and dilutions used are listed in Supplementary Table 3. After 3 washes, the membrane was incubated with secondary antibody conjugated to horseradish peroxidase for 1 h at room temperature. After 3 washes, signal was visualized by chemiluminescence using the Luminata Forte substrate (Thermo Fisher Scientific) and images were acquired with the ChemiDoc™ Touch Imaging System (Bio-Rad, Hercules, CA). When blotting of a single membrane for different proteins was necessary, the membrane was stripped using the Restore PLUS Stripping Buffer (Thermo Fisher Scientific) according to producer's instructions and the immunoblotting process was repeated.

**RNA extraction from cells, RNA sequencing and analysis, qPCR**. Total RNA was extracted from cells using the Maxwell 16 LEV simplyRNA Purification Kit and the Maxwell 16 Instrument. RNA integrity was verified using the Agilent Bioanalyzer 2100 (Agilent Technologies). cDNA was synthesized from total RNA using Superscript III (Invitrogen). Library preparation was performed using TruSeq RNA Library Preparation Kit v2. RNA sequencing was performed on the HiSeq 2500 sequencer to generate 2 × 75 bp paired-end reads.

Sequence reads were aligned using STAR two-pass[95] to the human reference genome GRCh37. Gene counts were quantified using the "GeneCounts" option. Per-gene counts-per-million (CPM) were computed and $\log_2$-transformed adding a pseudo-count of 1 to avoid transforming 0. Genes with $\log_2$-CPM <1 in more than three samples were removed. Unsupervised clustering was performed using the top 500 most variable genes, Euclidean distance as the distance metric and the Ward clustering algorithm. When required, the batch effect was removed using the function removeBatchEffect from the limma R package for data visualization. For differential expression the batch factor was included in the design matrix.

Differential expression analysis between knock-down cells and control samples was performed using the edgeR v3.28.1 package[96]. Normalization was performed using the "TMM" (weighted trimmed mean) method and differential expression was assessed using the quasi-likelihood F-test.

Genes with FDR < 0.05 and >2-fold were considered significantly differentially expressed.

Gene Set Enrichment Analysis (GSEA) was performed using the Preranked tool[97] for C2 (canonical pathways) and H (hallmark gene sets)[98]. Genes were ranked based on the T-statistic from the differential expression analysis.

Primer sequences used for qPCR are available in Supplementary Table 5.

**Cell growth experiments**. Cells were treated with siRNA (3 pmol) against *SMARCA4, SMARCA2, SMARCC1, SMARCC2* or with a scrambled control for 24 h. LNCaP and C4-2 cells were then seeded in Poly-L-Lysine coated 96-well plates (2000 cells/well) and WCM154 cells were seeded in a collagen-coated 96-well plates (5000 cells/well). Cell viability was determined after 24, 48, 72, and 96 h with a Tecan Infinite M200PRO reader using the CellTiter-Glo® Luminescent Cell Viability Assay according to manufacturer's directions (Promega). Cell confluence was determined using the Incucyte S3 instrument and the IncuCyte S3 2018B software (Essen Bioscience, Germany). Values were calculated as *x*-fold of cells transfected with siRNA for 0 h.

**Co-immunoprecipitation and mass spectrometry analysis**. For the co-immunoprecipitation (co-IP) using an anti-BAF155 antibody, nuclear fractions of LNCaP-AR and NCI-H660 cells were isolated using the using the Universal CoIP Kit (Actif Motif). Chromatin of the nuclear fraction was mechanically sheared using a Dounce homogenizer. Nuclear membrane and debris were pelleted by centrifugation and protein concentration of the cleared lysate was determined with the Pierce BCA Protein Assay Kit (Thermo Fisher Scientific). 2 μg of the anti-BAF155 antibody (ab172638, Abcam) and 2 μg of rabbit IgG Isotype Control antibody (026102, Thermo Fisher Scientific) were incubated with 2 mg protein supernatant overnight at 4 °C with gentle rotation. The following morning, 30 μl of Protein G Magnetic Beads (Active Motif) were washed twice with 500 μl CoIP buffer and incubated with Antibody-containing lysate for 1 h at 4 °C with gentle rotation. Bead-bound SWI/SNF complexes were washed 3 times with CoIP buffer and twice with a buffer containing 150 mM NaCl, 50 mM Tris-HCL (pH 8) and Protease and Phosphatase inhibitors. Air-dried and frozen (−20 °C) beads were subjected to mass spectrometry (MS) analysis. Proteins on the affinity pulldown beads were re-suspended in 8 M Urea/50 mM Tris-HCl pH 8, reduced 30 min at 37 °C with DTT 0.1 M/100 mM Tris-HCl pH 8, alkylated 30 min at 37 °C in the dark with IAA 0.5 M/100 mM Tris-HCl pH 8, diluted with 4 volumes of 20 mM Tris-HCl pH 8/2 mM $CaCl_2$ prior to overnight digestion at room temperature with 100 ng sequencing grade trypsin (Promega). Samples were centrifuged and the magnetic beads trapped by a magnet holder in order to extract the peptides in the supernatant.

The digests were analyzed by liquid chromatography (LC)-MS/MS (PROXEON coupled to a QExactive HF mass spectrometer, ThermoFisher Scientific) with three injections of 5 μl digests. Peptides were trapped on a μPrecolumn C18 PepMap100 (5 μm, 100 Å, 300 μm × 5 mm, ThermoFisher Scientific, Reinach, Switzerland) and

separated by backflush on a C18 column (5 μm, 100 Å, 75 μm × 15 cm, C18) by applying a 60-min gradient of 5% acetonitrile to 40% in water, 0.1% formic acid, at a flow rate of 350 nl/min. The Full Scan method was set with resolution at 60,000 with an automatic gain control (AGC) target of 1E06 and maximum ion injection time of 50 ms. The data-dependent method for precursor ion fragmentation was applied with the following settings: resolution 15,000, AGC of 1E05, maximum ion time of 110 ms, mass window 1.6 *m/z*, collision energy 28, under fill ratio 1%, charge exclusion of unassigned and 1+ ions, and peptide match preferred, respectively.

MS data was interpreted with MaxQuant (version 1.6.1.0) against a SwissProt human database (release 2019_02) using the default MaxQuant settings, allowed mass deviation for precursor ions of 10 ppm for the first search, maximum peptide mass of 5500 Da, match between runs activated with a matching time window of 0.7 min and the use of non-consecutive fractions for the different pulldowns to prevent over-fitting. Settings that differed from the default setting included: strict trypsin cleavage rule allowing for 3 missed cleavages, fixed carbamidomethylation of cysteines, variable oxidation of methionines and acetylation of protein N-termini.

Protein intensities are reported as MaxQuant's Label Free Quantification (LFQ) values, as well as Top3 values (sum of the intensities of the three most intense peptides); for the latter, variance stabilization was used for the peptide normalization, and missing peptide intensities were imputed in the following manner: if there was at least two evidences in one group of replicates, the missing value was drawn from a Gaussian distribution of width 0.3 centered at the sample distribution mean minus 1.8× the sample standard deviation. Imputation at protein level for both LFQ and Top3 values was performed if there were at least two measured intensities in at least one group of replicates; missing values in this case were drawn from a Gaussian distribution of width 0.2 centered at the sample distribution mean minus 2.5x the sample standard deviation. Differential expression tests were performed using the moderated *t*-test *empirical Bayes* (R function EBayes from the limma package version 3.40.6) on imputed LFQ and Top3 protein intensities. The Benjamini and Hochberg method was further applied to correct for multiple testing. The criterion for statistically significant differential expression is that the maximum adjusted *p*-value for large fold changes is 0.05, and that this maximum decreases asymptotically to 0 as the log2 fold change of 1 is approached (with a curve parameter of one time the overall standard deviation).

Please see below, the description of the methods for the second Co-IP (validation experiment) using an anti-BRG1 antibody in LNCaP and NCI-H660 cells.

**CRISPR-Cas9 mediated *TP53* and *RB1* knock-out**. To generate the stable p53 and RB1 knockout cells, all-in-one CRISPR plasmids with mCherry reporter were purchased from Genecopoeia (Cat # HCP218175-CG01, HCP216131-CG01). Cells were transfected with CRISPR plasmids, selected with puromycin and sorted for mCherry positivity. *TP53* gRNA sequences used: TCGACGCTAGGATCTGACTG, CGTCGAGCCCCCTCTGAGTC, CCATTGTTCAATATCGTCCG. *RB1* gRNA sequences used: CGGTGGCGGCCGTTTTTCGG, CGGTGCCGGGGGGTTCCGC GG, CGGAGGACCTGCCTCTCGTC. Control gRNA sequence: GGCTTCGCGCC GTAGTCTTA.

**Single cell RNA-sequencing (scRNAseq)**. scRNAseq was performed for two CRPC-NE organoids in 3D culture: MSK PCa1 and MSK PCa16. Cell counting and viability tests were conducted using a Moxi Go II Flow Cytometer (Orflo Technologies) with trypan blue and Propidium Iodide staining. Subsequently, GEM generation and barcoding, reverse transcription, cDNA amplification and 3′ Gene Expression library generation steps were all performed according to the Chromium Single Cell 3′ Reagent Kits v3 user Guide (10× Genomics CG000183 Rev B). Specifically, 32.0, 11.4, and 40.0 μL of PCa1, PCa8, and PCa16 cell suspension (100, 750, and 200 cells/μL) were used for a targeted cell recovery of 2000, 5000, and 2000 cells, respectively. GEM generation was followed by a GEM-reverse transcription incubation, a clean-up step and 12 cycles of cDNA amplification. The resulting cDNA was assessed for quantity and quality using fluorometry and capillary electrophoresis, respectively. The cDNA libraries were pooled and sequenced paired-end and single indexed on an illumina NovaSeq 6000 sequencer with a shared NovaSeq 6000 S2 Reagent Kit (100 cycles). The read-set up was as follows: read 1 = 28 cycles, i7 index = 8 cycles, i5 = 0 cycles and read 2 = 91 cycles. An average of 300,753,777 reads/library were obtained, equating to an average of 111, 978 reads/cell. All steps were performed at the Next Generation Sequencing Platform, University of Bern. Data demultiplexing was performed using SEURAT v. 3.1.5 package (PMID 29608179). Low quality cells and multiplets were excluded by removing cells with unique feature counts over 5500 or less than 1000. Cells containing mitochondrial gene counts greater than 25% were also removed. Data were then scaled to 10,000 and log transformed. Only cells expressing *SOX2* and *SMARCA4* genes were included. Boxplots were drawn using GGPLOT2 3.3.0 (https://ggplot2.tidyverse.org) and *p*-values were calculated using Wilcoxon test.

**ChIP-sequencing data analysis**. ChIP-seq peaks for *SMARCC1* and *HOXB13* in LNCaP cells were downloaded from GEO: GSE110655 and GSE94682, respectively. ChIP-seq peaks for H3K27ac and H3K27me3 in LNCaP cells were from data

published by Sandoval et al.[71]. Peak comparison was performed using BEDTOOLS v2-29.0 (https://bedtools.readthedocs.io/en/latest/#).

**Co-immunoprecipitation using the anti-SMARCA4 antibody and mass spectrometry analysis**. For the second Co-IP (validation experiment) using an anti-SMARCA4 antibody (results shown in Supplementary Fig. 26 and Supplementary Data 9), SWI/SNF complexes were isolated from the nuclear fraction of LNCaP (adenocarcinoma) or NCI-H660 (CRPC-NE) cells, which was prepared using the Universal CoIP Kit (Active Motif). Briefly, anti-Brg-1 antibodies (H-10, Santa Cruz Biotechnology) were cross-linked using Dimethyl pimelimidate dihydrochloride (Sigma-Aldrich) to Protein G conjugated magnetic beads (Bio-Rad). 30 μg of cross-linked antibodies were incubated with 0.8–1 mg of nuclear lysates overnight. Bead-bound BAF complexes were washed and eluted using 8 M urea buffer. The obtained protein complexes were subjected to immunoblotting and MS analysis.

For MS analysis, the eluted proteins were precipitated with trichloroacetic acid (TCA, 20% w/v), rinsed three times with acetone, and dried at room temperature. The pellets were re-suspended in 50 μL resuspension buffer (8 M urea, 50 mM ammonium bicarbonate, and 5 mM DTT) and subjected to reduction and alkylation by adding 15 mM iodoacetamide to each sample for 30 min in the dark at room temperature, followed by addition of 5 mM DTT to quench the reaction. Samples were diluted to a final concentration of 2 M urea and digested with LysC at room temperature overnight, and then diluted further to 1 M urea and digested with Trypsin at 37 °C overnight (for each enzyme a ratio of 1:125 enzyme:protein was used).

Samples were labeled using reductive dimethylation. Labeling was done while the peptides were bound to the solid phase C18 resin in self-packed STAGE Tip micro-columns[99]. Stage tips were washed with methanol, acetonitrile (ACN) 70% v/v and formic acid (FA) 1% v/v. Samples were acidified by adding 100% FA to a final concentration of 2% FA before loading. After sample loading, stage tips were washed with 1% FA and phosphate/citrate buffer (0.23 M sodium phosphate and 86.4 mM citric acid [pH 5.5]). At this point, the "light" solution (0.4% $CH_2O$ and 60 mM $NaBH_3CN$), or "heavy" solution (0.4% $CD_2O$ and 60 mM $NaBD_3CN$) was added twice on each stage tip to label the peptides. A final wash with 1% FA was performed prior to elution with 70% ACN and 1% FA. Samples were dried under vacuum, resuspended in 5% FA, and mixed together in equal amounts for analysis using an Orbitrap Fusion Mass Spectrometer. Peptides were introduced into the mass spectrometer by nano-electrospray as they eluted off a self-packed 40 cm, 75 μm (ID) reverse-phase column packed with 1.8 μm, 120 Å pore size, SEPAX C18 resin. Peptides were separated with a gradient of 5–25% buffer B (99.9% ACN, 0.1% FA) with a flow rate of 350 nl/min for 65 min. For each scan cycle, one high mass resolution full MS scan was acquired in the Orbitrap mass analyzer at a resolution of 120 K, AGC value of 500,000, in a *m/z* scan range of 375–1400, max acquisition time of 100 ms and up to 20 parent ions were chosen based on their intensity for collision induced dissociation (normalized collision energy = 35%) and MS/MS fragment ion scans at low mass resolution in the linear ion trap. Dynamic exclusion was enabled to exclude ions that had already been selected for MS/MS in the previous 40 s. Ions with a charge of +1 and those whose charge state could not be assigned were also excluded. All scans were collected in centroid mode. Two biological replicates for each condition were processed and analyzed.

MS2 spectra were searched using SEQUEST (version 28 revision 13) against a composite database containing all Swiss-Prot reviewed human protein sequences (20,193 target sequences, downloaded from www.uniprot.org March 18, 2016) and their reversed complement, using the following parameters: a precursor mass tolerance of ±25 ppm; 1.0 Da product ion mass tolerance; tryptic digestion; up to two missed cleavages; static modifications of carbamidomethylation on cysteine (+57.0214) and dimethylation on n-termini and lysines (+28.0313); dynamic modifications of methionine oxidation (+15.9949) and heavy dimethylation on N-termini and lysines (+6.03766). Peptide spectral matches (PSMs) were filtered to 1% FDR using the target-decoy strategy[100] combined with linear discriminant analysis (LDA)[101] using several different parameters including Xcorr, ΔCn', precursor mass error, observed ion charge state, and predicted solution charge state. Linear discriminant models were calculated for each LC-MS/MS run using peptide matches to forward and reversed protein sequences as positive and negative training data. PSMs within each run were sorted in descending order by discriminant score and filtered to a 1% FDR as revealed by the number of decoy sequences remaining in the data set. The data were further filtered to control protein level FDRs. Peptides were combined and assembled into proteins. Protein scores were derived from the product of all LDA peptide probabilities, sorted by rank, and filtered to 1% FDR as described for peptides. The FDR of the remaining peptides fell dramatically after protein filtering. The data were further filtered to require a minimum of 8 PSMs per protein. All peptides were required to have a sum of heavy and light signal-to-noise (SN) ≧10. Protein ratios were calculated as the $\log_2$ ratio of the total SN of all experimental sample peptide values over that for IgG control sample peptides. For a small number of the most highly enriched proteins, the control value was zero (this is the theoretical ideal). In these cases, we imputed a value of one for ratio calculations. Subsequent visualization and statistical analysis was done with Perseus and R program[102].

**SMARCC1 Co-IP immunoblotting**. For the co-immunoprecipitation (co-IP), using an anti-BAF155 antibody, nuclear fractions of LNCaP-AR and NCI-H660 cells were isolated using the using the Universal CoIP Kit (Active Motif). Chromatin of the nuclear fraction was mechanically sheared using a Dounce homogenizer. Nuclear membrane and debris were pelleted by centrifugation and protein concentration of the cleared lysate was determined with the Pierce BCA Protein Assay Kit (Thermo Fisher Scientific). One microgram of the rabbit anti-BAF155 antibody (ab172638, Abcam) and 1 μg of rabbit IgG Isotype Control antibody (026102, Thermo Fisher Scientific) were incubated with 1 mg protein supernatant overnight at 4 °C with gentle rotation. The following morning, 30 μl of Protein G Magnetic Beads (Active Motif) were washed twice with 500 μl CoIP buffer and incubated with Antibody-containing lysate for 2 h at 4 °C with gentle rotation. Bead-bound SWI/SNF complexes were washed twice with CoIP buffer and twice with a buffer containing 150 mM NaCl, 50 mM Tris-HCL (pH 8) and Protease and Phosphatase inhibitors. Washing procedure was executed at 4 °C with gentle rotation. Bead-bound protein and Input controls are reduced and denatured in 40 μl Laemmli buffer containing DTT through boiling for 5 min at 95 °C. Magnetic beads are removed from solution and 20 μl of reduce protein is loaded on an SDS-PAGE gel with subsequent immunoblotting using iBlot (Life Technologies). Membranes were blocked in 5% dry-milk solution and then incubated over night with respective antibodies against targets of interest. Protein signal was detected using HRP-labeled native anti-rabbit IgG antibody (CST, #5127) and ECL substrate solution (Merck Millipore) using the Fusion FX.

**RNA isolation and qPCR**. Cells were first seeded in 10cm-petridish and grown until they reached a confluency of approx. 90%. The cells were then harvested for RNA isolation using the RNeasy Mini Kit (Qiagen). Synthesis of complementary DNAs (cDNAs) using FIREScript RT cDNA Synthesis Kit (Solis BioDyne) and real-time reverse transcription PCR (RT-PCR) assays using HOT FIREPol Eva-Green qPCR Mix Plus (Solis BioDyne) were performed using and applying the manufacturer protocols. Relative mRNA levels of each gene shown were normalized to the expression of the average of housekeeping genes GAPDH and ACTB. The sequences of the primers for qRT-PCR assays can be found in Supplementary Table 5.

**Reporting summary**. Further information on research design is available in the Nature Research Reporting Summary linked to this article.

## Data availability

Data generated during this study have been submitted on the European Genome-phenome Archive under the accession EGAS00001004177 (https://ega-archive.org/datasets/EGAD00001005800). The mass spectrometry proteomics data that support the findings of this study have been deposited to the ProteomeXchange Consortium (http://proteomecentral.proteomexchange.org) via the PRIDE partner repository with the dataset identifier PXD016861. Source data are provided with this paper.

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

## Acknowledgements

We thank patients and their families for participating in genomics, transcriptomics, and precision cancer care studies. The authors would like to thank Cigall Kadoch from Dana Farber Cancer Institute for her important insights into SWI/SNF biology. We thank current and previous members of the Rubin lab for valuable discussions, Charles Sawyers and Ping Mu at Memorial Sloan-Kettering Cancer Center for sharing the LNCaP-AR cell line. We are grateful to Juan Miguel Mosquera, Brian Robinson, and Verena Sailer for their prostate cancer tissue contributions. We are thankful for expert assistance from the Translational Research Program at WCMC Pathology and Laboratory Medicine (Bing He, Leticia Dizon, Yifang Liu, Mai Ho) and the WCM CLC Genomics Core Facility (Jenny Xiang). We thank Inti Zlobec and Micha D. Eichmann at the University of Bern for their assistance with the Scorenado TMA scoring platform. We thank Elai Davicioni (Decipher Biosciences, Cam, USA) for providing data for gene expression analyses. We further thank the Next Generation Platform (Institute of Genetics, Bern) and Irene Keller (DBMR, Bern) for their assistance with the single cell experiments. We acknowledge expert assistance from Mariana Ricca at the University of Bern in preparing the manuscript for submission. This project has received funding from the Nuovo-Soldati Foundation (J.C.), the NIH/NCI WCM SPORE in Prostate Cancer P50-CA211024 (V.C., H.B., K.B., F.D., M.A.R.), the Swiss National Science Foundation Project grants number 310030_189149 and 31003A_169352 (M.K.-d.J.) and Ambizione grant number: PZ00P3_168165 (S.P.), the NCI grants number 1R01CA233650 (P.C.), 1R01CA233650 (N.D.), R37 CA241486 (H.B.), and R01 CA125612 (F.D.) and the European Research Council (ERC) under the European Union's Horizon 2020 research and innovation programme (grant agreement No 648670) (F.D.).

## Author contributions

J.C., A.A., and M.A.R. designed the study and the experiments. J.C., A.A., P.T., D.W., S.-S.C., S.C., M.J., and L.B. performed experiments. M.R.d.F., S.P., D.P., M.B., and F.D. performed genomic and transcriptomic analyses of patient data. M.R.d.F., S.P., R.B., and A.S. performed RNA-seq analysis of experimental data. L.R. and S.P. performed scRNA-seq and ChIP-seq analysis. M.R.d.F., S.P., and M.A. performed analyses on Decipher GRID and JHMI cohorts. V.C. and K.V.B. performed survival analyses. J.C. and M.A.R. performed pathology review and immunohistochemical evaluation. P.C. and N.D. performed mass spectrometry experiments and analyzed respective data. A.-C.U., S.B.L. performed mass spectrometry analyses. F.F., T.L.L., M.S. and M.K.-d.J. established annotated patient cohorts and provided clinical data. H.B. provided CRPC-NE patient samples, L.P., S.W., and Y.C. established patient-derived organoid models. M.A.R. provided administrative, technical and material support. J.C., A.A., and M.A.R. wrote the initial draft of the manuscript and all authors contributed to the final version.

## Competing interests

H.B. has received research funding from Janssen, Astellas, Abbvie, Millennium, and Eli Lilly and consulting with Janssen, Astellas, Sanofi Genzyme, Astra Zeneca, Pfizer. L.P. is now an employee of Loxo Oncology at Lilly. M.A.R. has received research funding from Novartis, Roche, Ventana, Janssen, Astellas, Millennium, and Eli Lilly. M.A.R. is on the SAB of Neogenomics. T.L.L. has received research funding from DeepBio, Decipher, Ventana/Roche. Cornell and Bern Universities have filed a patent application on SWI/SNF diagnostic and therapeutic fields with A.A., J.C., and M.A.R. listed as inventors. The remaining authors declare no competing interests.

## Additional information

Joanna Cyrta[1,2,29], Anke Augspach[1,29], Maria Rosaria De Filippo[3,4], Davide Prandi[5], Phillip Thienger[1], Matteo Benelli[5,6], Victoria Cooley[7], Rohan Bareja[2,8], David Wilkes[2], Sung-Suk Chae[9], Paola Cavaliere[10], Noah Dephoure[10,11], Anne-Christine Uldry[12], Sophie Braga Lagache[12], Luca Roma[4], Sandra Cohen[9], Muriel Jaquet[1], Laura P. Brandt[1], Mohammed Alshalalfa[13], Loredana Puca[14], Andrea Sboner[2,8,15,16], Felix Feng[12], Shangqian Wang[17], Himisha Beltran[14,18], Tamara Lotan[19,20,21], Martin Spahn[22,23], Marianna Kruithof-de Julio[1,3,24], Yu Chen[14], Karla V. Ballman[7], Francesca Demichelis[2,5], Salvatore Piscuoglio[4,25,26,30] & Mark A. Rubin[1,27,28,30]✉

[1]Department for BioMedical Research, University of Bern, 3008 Bern, Switzerland. [2]The Caryl and Israel Englander Institute for Precision Medicine, Weill Cornell Medicine, New York, NY 10021, USA. [3]Department for BioMedical Research, Urology Research Laboratory, University of Bern, 3008 Bern, Switzerland. [4]Institute of Pathology and Medical Genetics, University Hospital Basel, University of Basel, 4051 Basel, Switzerland. [5]Department of Cellular, Computational and Integrative Biology (CIBIO), University of Trento, 38122 Trento, Italy. [6]Bioinformatics Unit, Hospital of Prato, 59100 Prato, Italy. [7]Department of Healthcare Policy and Research, Division of Biostatistics and Epidemiology, Weill Cornell Medicine, New York, NY 10021, USA. [8]Institute for Computational Biomedicine, Weill Cornell Medicine, New York, NY 10021, USA. [9]Department of Laboratory Medicine and Pathology, Weill Cornell Medicine, New York, NY 10021, USA. [10]Meyer Cancer Center, Weill Cornell Medicine, New York, NY 10021, USA. [11]Department of Biochemistry, Weill Cornell Medicine, New York, NY 10021, USA. [12]Proteomics Mass Spectrometry Core Facility, University of Bern, 3010 Bern, Switzerland. [13]Department of Radiation Oncology, Helen Diller Family Comprehensive Cancer Center, University of California at San Francisco, San Francisco, CA, USA. [14]Department of Medicine, Division of Medical Oncology, Weill Cornell Medicine, New York, NY 10021, USA. [15]HRH Prince Alwaleed Bin Talal Bin Abdulaziz Alsaud Institute for Computational Biomedicine, Weill Cornell Medicine, New York, NY 10021, USA. [16]Meyer Cancer Center, Weill Cornell Medicine, New York, NY 10065, USA. [17]Human Oncology and Pathogenesis Program and Department of Medicine, Memorial Sloan-Kettering Cancer Center, New York, NY 10065, USA. [18]Department of Medical Oncology, Dana Farber Cancer

Institute, Boston, MA, USA. [19]Department of Urology, Johns Hopkins University School of Medicine, Baltimore, Maryland, USA. [20]Department of Pathology, Johns Hopkins University School of Medicine, Baltimore, MD 21205, USA. [21]Department of Oncology, Johns Hopkins University School of Medicine, Baltimore, MD 21205, USA. [22]Lindenhofspital Bern, Prostate Center Bern, 3012 Bern, Switzerland. [23]Department of Urology, Essen University Hospital, University of Duisburg-Essen, 47057 Essen, Germany. [24]Department of Urology, Inselspital, 3010 Bern, Switzerland. [25]Visceral Surgery Research Laboratory, Clarunis, Department of Biomedicine, University of Basel, 4051 Basel, Switzerland. [26]Clarunis Universitäres Bauchzentrum Basel, 4002 Basel, Switzerland. [27]Inselspital, 3010 Bern, Switzerland. [28]Bern Center for Precision Medicine, 3008 Bern, Switzerland. [29]These authors contributed equally: Joanna Cyrta, Anke Augspach. [30]These authors jointly supervised this work: Salvatore Piscuoglio, Mark A. Rubin. ✉email: mark.rubin@dbmr.unibe.ch

