## [Peer Review File · Nature Communications]

Reviewers' comments:

Reviewer #1 (Remarks to the Author):

The manuscript by Joanna Cyrt et al entitled "Role of specialized composition of SWI/SNF complexes in prostate cancer lineage plasticity" investigated the role of SWI/SNF complex in CRPC-NE. The authors demonstrated a role of upregulated BRG1 in CRPC-NE compared with other PCa. Interestingly, using publicly available databases, the authors showed that BRG1 overexpression is associated with aggressive disease. Finally, the authors show that SWI/SNF complexes interact with different lineage-specific factors in CRPC-NE compared with adenocarcinoma of prostate. Although SWI/SNF complexes including BRG1 have been studied in prostate cancer, the present study focuses on CRPC-NE is considered novel. Overall, the manuscript is interesting and experimental design is appropriate and well controlled. Thus, the manuscript may be appropriate for Nature Communications pending addressing the following comments:

Major comments:

- 1) The switch between BRG1 and BRM in NEPC is interesting. The authors performed BRG1 and BRM knockdown, it would be important to perform BRG1 overexpression in non-CRPC-NE cells to determine whether BRG1 overexpression is sufficient to drive the NEPC phenotype and upregulation of the BRG1-knockdown gene expression signature.
- 2) While the interaction with lineage-specific factors is intriguing, the authors failed to demonstrate the functional significance of these interactions. For example, the changes in genome-wide association of the SWI/SNF complexes and the association with the reported changes in gene expression signature (direct vs. indirect) would be critical for the full appreciation of the importance of the observed changes.
- 3) It is informative to examine the response to ARSi in CRPC-NE and non-CRPC-NE cells knockdown and/or overexpression of BRG1 and BRM to establish the mechanistic link between database mining and biological consequences.
- 4) Given the discrepancy between the present and previous studies, it is informative to examine whether BRG1 knockdown suppresses the growth of PTEN-competent CRPC-NE cells.
- 5) Since EZH2 is overexpressed in CRPC-NE and regulated by BRG1, it is informative to examine whether EZH2 mediates the observed role of BRG1 in CRPC-NE.
- 6) Although the authors described the low overall rate of SWI/SNF mutation in all cases in Fig. 1, the number of CRPC-NE cases is small in the overall cohort. It might be productive to examine the rate of SWI/SNF alterations only in CRPC-NE cases. For example, the authors described a significant difference in the rates of BRD7, PBRM1 and SMARCD1 mutation between CRPC-NE and CRPC-Adeno. However, there is no follow-up on these mutations to determine whether they are important functionally.
- 7) Given the mutual exclusivity between BRG1 and BRM, it is odd to see BRG1 pulled down BRM in Fig. S1.6. Are they in the same complex or this is due to high background in IP analysis?
- 8) In Fig. S1.7, does BRG1 knockdown in CRPC-NE cells affect neuronal genes?
- 9) Given the role of REST in regulating BAF53B and neuronal genes, it is informative to examine whether BRG1 knockdown affects REST expression.
- 10) Expression status of BRG1 and BRM should be included in Fig. 1e and BRM expression should be included in Fig. 4d.

Reviewer #2 (Remarks to the Author):

This study by Rubin and coworkers investigated the role of the mammalian SWI/SNF chromatin-remodeling complex in neuroendocrine prostate cancer. This is achieved through integrating a large collection of clinical prostate cancer datasets with some functional studies in organoids and

cell lines. The results show that mammalian SWI/SNF subunits are deregulated and indicate tumor-promoting roles in neuroendocrine prostate cancer. Consistent with several previous studies, high SMARCA4 expression was observed to be associated with aggressive types of prostate cancers. Finally, the authors report that the SWI/SNF complexes interact with different lineage-specific factors in neuroendocrine prostate cancer compared to prostate adenocarcinoma.

Overall, this study is well carried out and is very complete in terms of clinical correlation analysis of the SWI/SNF complex performed. The following suggestions are for improvement.

1. In Figure 1b, compared to CRPC-Adeno, a significant increase of LOH fraction in CRPC-NE was observed for three genes: BRD7, SMARCD1, and PBRM1. The authors may provide gene expression data to show whether LOH events correlate with their transcript levels.
2. The results in Figure 1 clearly showed that the SMARCA4 was significantly upregulated while SMARCA2 was downregulated at both transcriptional and protein levels in CRPC-NE. For characterizing the effects of SMARCA4 and SMARCA2, why the authors did not use CRPC-NE organoid model as that of the other SWI/SNF subunits BAF53B and BAF45B? NE-like PC3 cell line (PMID: 21432867) may be a good model for this purpose. Meanwhile, it would be interesting to know to what degree the impact of SMARCA4 knock-down while overexpressing SMARCA2 in CRPC-NE cell lines, and organoids if possible. To solidify the findings, in addition to siRNA-mediated knock-down for SMARCA4 and SMARCA2, the complementary assays using shRNA and/or CRISPR-Cas9 are encouraged.
3. What is also missing is to analyze transcriptional and epigenetic reprogramming in CRPC-NE versus prostate adenocarcinoma cells while depleting the core SWI/SNF subunit BRG1 or SMARCC1. In parallel RNA-seq and chromatin profiling such as ATAC-seq may be useful to address lineage-specific roles of the SWI/SNF complex corroborating with their lineage-specific interaction partners in particular transcription factors.

Minor comments:

1. Minor errors, such as 'previously described³⁰' in line 4 page 21, 'Hochberg method⁹⁹' in line 4 page 27; 'and in CRPC-Adeno versus CRPC-Adeno (c)' in Supplementary Figure S1.1, two 'Lowest' in Supplementary Figure S3.2b.
2. It is likely to miss proper references in line 1 page 10 "significant decrease in PCa cell growth, in line with previous studies".

NCOMMS-20-05343-T: Point by Point Response

Reviewers' comments:

Reviewer #1 (Remarks to the Author):

The manuscript by Joanna Cyrta et al entitled "Role of specialized composition of SWI/SNF complexes in prostate cancer lineage plasticity" investigated the role of SWI/SNF complex in CRPC-NE. The authors demonstrated a role of upregulated BRG1 in CRPC-NE compared with other PCa. Interestingly, using publicly available databases, the authors showed that BRG1 overexpression is associated with aggressive disease. Finally, the authors show that SWI/SNF complexes interact with different lineage-specific factors in CRPC-NE compared with adenocarcinoma of prostate. Although SWI/SNF complexes including BRG1 have been studied in prostate cancer, the present study focuses on CRPC-NE is considered novel. Overall, the manuscript is interesting and experimental design is appropriate and well controlled. Thus, the manuscript may be appropriate for Nature Communications pending addressing the following comments:

Response: We thank the Reviewer for these thoughtful and supportive comments.

Major comments:

1) The switch between BRG1 and BRM in NEPC is interesting. The authors performed BRG1 and BRM knockdown, it would be important to perform BRG1 overexpression in non-CRPC-NE cells to determine whether BRG1 overexpression is sufficient to drive the NEPC phenotype and upregulation of the BRG1-knockdown gene expression signature.

Response #1. This is a great point. We tried to address this question in a prior experiment. 22Rv1 prostate adenocarcinoma cells were lentiviral transduced to overexpress either SMARCA4 or SMARCA2 (pEZ-Lv203 or pEZLv216 vectors, respectively; Genecopoeia, MD, USA), or with a matched empty control vector, and sorted based on the expression of the fluorescent reporter (eGFP or mCherry, respectively). Despite strong expression of the fluorescent reporters, we did not observe an increase in SMARCA4 or SMARCA2 at protein level. However, we observed an overexpression of SMARCA4 and SMARCA2 in transduced cells after an additional 24h treatment with the proteasome inhibitor MG-132, as shown in the figure below. These findings hint towards a tight and cellular context-dependent regulation of catalytic SWI/SNF subunits. Forced, isolated overexpression of one subunit may provoke rapid degradation of this excess subunit. Thus, we hypothesize that SMARCA4 overexpression may be necessary but not sufficient to promote an aggressive phenotype in prostate cancer cells.

Changes to manuscript #1. We have now included the above figure as a **Supplemental Fig. S2.9** and have added a sentence in the Results section to report these findings.

2) While the interaction with lineage-specific factors is intriguing, the authors failed to demonstrate the functional significance of these interactions. For example, the changes in genome-wide association of the SWI/SNF complexes and the association with the reported changes in gene expression signature (direct vs. indirect) would be critical for the full appreciation of the importance of the observed changes.

Response #2. We thank the Reviewer for these valuable suggestions. The aim of this study was to provide a general overview of SWI/SNF deregulation in the context of CRPC-NE. As such, exploring the functional impact of the observed interactions with lineage-specific factors is considered beyond the scope of this manuscript but will definitely be followed up in future experiments and publications.

HOXB13 is one of the transcription factors that we found to be specifically associated with SWI/SNF in prostatic adenocarcinoma cells. To further explore the possibility of a functional interaction between HOXB13 and SWI/SNF, we performed a proof-of-principle computational analysis of published ChIP-seq datasets for SMARCC1, HOXB13, the inactive chromatin histone mark H3K27me3 and the active chromatin histone mark H3K27ac in LNCaP cells. We observed an important overlap between SMARCC1 and HOXB13 at active chromatin sites (11,824 sites). Conversely, there was almost no overlap between the chromatin occupancy sites common to SMARCC1 and HOXB13 at inactive chromatin sites. These observations further support the hypothesis that the interaction between SWI/SNF and lineage-specific factors in PCa may be meaningful at the functional level.

New Supplemental Fig. 4.4. Venn diagrams illustrating the overlap in genome occupancy sites for SMARCC1, HOXB13, H3K27me3 and H3K27ac in LNCaP cells, assessed by ChIP-seq.

Changes to manuscript #2. This result has been added to the Results section of the manuscript and included as **Supplemental Fig. 4.4.**

3) It is informative to examine the response to ARSi in CRPC-NE and non-CRPC-NE cells knockdown and/or overexpression of BRG1 and BRM to establish the mechanistic link between database mining and biological consequences.

Response #3. This is indeed an important point in the context of prostate cancer and we fully agree that it to verify whether BRG1 knockdown in CRPC-NE can restore sensitivity to ARSi. Unfortunately, given the current situation, we are unable to perform these additional experiments in a timely manner. Nevertheless, we did query the potential effect of BRG1 and BRM knock-down on AR signaling. siRNA-mediated BRG1 knock-down was performed in LNCaP-AR cells stably transduced with a doxycycline-inducible shRNA vector allowing for simultaneous knock-down of TP53 and RB1 (a cell model kindly provided by Drs. Charles Sawyers and Ping Mu, and published in PMID 28059768). Gene expression levels were queried at 96h using the Nanostring system to assess the AR signaling score (Hieronymus et al., PMID 17010675). As shown in the figure below (triplicate experiments, paired two-tailed t test), in untreated cells, BRG1 knock-down led to a slight, albeit non-significant increase of the AR signaling score, as compared to the Scrambled control. In doxycycline-treated cells in which TP53/RB1 knock-down had been induced, the AR signaling score was overall lower than in untreated cells, as expected, but was significantly higher in cells with BRG1 knock-down than in the Scrambled control.

Lastly, we would like to point out the possibility of potential confounding factors in such analyses, including the fact that BRG1 loss strongly decreases growth of LNCaP cells and thus, could lead to non-specific increase of AR signaling, as a mechanism of cell response to stress. Overall, we feel that the relationship between BRG1, AR signaling and ARSi response should definitely be explored in future studies, but is beyond the scope of the current manuscript.

Changes to manuscript #3. No changes were made to the manuscript regarding this point. However, the above results and discussion points can be incorporated in the manuscript if the Editor and Reviewers feel it is warranted.

4) Given the discrepancy between the present and previous studies, it is informative to examine whether BRG1 knockdown suppresses the growth of PTEN-competent CRPC-NE cells.

Response #4. Thank you for raising this point. Ding et al. (PMID 30496141) found that BRG1 knock-down did not affect the growth of PTEN-competent cells. As such, our results are not discrepant with their data: we performed transient BRG1 knock-down in the CRPC-NE cell line WCM154 (described in Puca et al., PMID 29921838), a PTEN-competent cell line, and did not observe a significant negative but instead a positive effect on cell growth (figure below, left hand side).

We also performed transient knockdown of BRG1 and BRM in a PTEN-null CRPC-NE cell line, WCM155 (figure below, right hand side). There seems to be a trend towards a decrease in cell viability in the BRG1 knockdown condition but the result do not reach statistical significance. This might be due to the very low growth rate of this organoid cell line, and may require further studies using a stable and/or inducible knock-down system in order to measure timepoints longer than 96h. To expand upon findings by Ding et al., we also performed knock-down experiments of SMARCC1 and SMARCC2 in 22Rv1 cells and in WCM154 CRPC-NE cells (both PTEN-competent), and observed a significant growth reduction in both knock-down conditions (as shown in the figure below). This novel finding suggests that unlike the SWI/SNF components SMARCA4/A2, the SWI/SNF core components SMARCC1 and SMARCC2 may be essential for PCa cell growth not only PTEN-null, but also in PTEN-competent cells, including CRPC-NE cells.

Results of the viability assay (CellTiter-Glo, ratio versus T0) in WCM155 cells (CRPC-NE, PTEN-competent) upon siRNA-mediated knock-down of BRG1 or BRM. Error bars: SEM, two-way ANOVA test (result at 96h), ns: non-significant.

New Supplemental Fig. S2.7. Results of the growth assay (cell confluence, ratio of T0) in 22Rv1 cells and in WCM154 CRPC-NE cells (both PTEN-competent) upon siRNA-mediated knock-down of SMARCC1 or SMARCC2. Three pooled independent experiments are shown. Error bars: SEM, two-way ANOVA test (result at 120h and at 96h), ** $p < 0.005$, *** $p < 0.0005$.

Changes to manuscript #4. The results of SMARCC1 and SMARCC2 knock-down in 22Rv1 cells have been added to the Results section and included as **Supplemental Fig. S2.7**.

5) Since EZH2 is overexpressed in CRPC-NE and regulated by BRG1, it is informative to examine whether EZH2 mediates the observed role of BRG1 in CRPC-NE.

Response #5. This is a great suggestion. To address this, we compared the transcriptomic effects of BRG1 knock-down (RNA-seq data from our study) and of EZH2 knock-down (published data) in LNCaP cells. As shown in the figure below, we observed a significant enrichment of gene sets related to EZH2 inactivation, further supporting a functional overlap between BRG1 and EZH2. Of note, although we report a decrease in EZH2 expression upon BRG1 knock-down in prostate cancer cells at protein and transcript level (Current Fig. 2c and 2e), we cannot exclude an indirect mechanism for this, rather than a direct regulation of EZH2 expression by BRG1. In particular, the decrease in cell proliferation induced by BRG1 knock-down could result in a “collapse” of various gene expression programs implicated in cell cycle regulation, which includes EZH2.

New Supplemental Fig. S2.3. GSEA in LNCaP cells with BRG1 knock-down and with EZH2-related gene sets: genes upregulated in PC3 prostate cancer cells after knockdown of EZH2 (left) and genes downregulated in PC3 cells after knockdown of EZH2 (right).

Changes to manuscript #5. These results have been added to the Results section of the manuscript and included as new **Supplemental Fig. S2.3.**

6) Although the authors described the low overall rate of SWI/SNF mutation in all cases in Fig. 1, the number of CRPC-NE cases is small in the overall cohort. It might be productive to examine the rate of SWI/SNF alterations only in CRPC-NE cases. For example, the authors described a significant difference in the rates of *BRD7*, *PBRM1* and *SMARCD1* mutation between CRPC-NE and CRPC-Adeno. However, there is no follow-up on these mutations to determine whether they are important functionally.

Response #6. We thank the Reviewer for this valuable comment. We tried to present the genomic data in Fig. 1 in a manner that would illustrate a break-down by disease stages. We apologize if this was not well highlighted in the text.

The significantly higher prevalence in the rates of *BRD7*, *PBRM1* and *SMARCD1* alterations in CRPC-NE (Fig. 1), refers to copy number alterations (heterozygous loss and copy-neutral LOH). To clarify whether these alterations may display functional significance, we verified the expression levels of these genes in CRPC-NE (as shown in the figure below). For *PBRM1* and *SMARCD1*, we did not observe a significant expression decrease in the CRPC-NE group. For *BRD7*, expression was significantly lower in CRPC-NE compared to CRPC-Adeno, but not in CRPC-NE compared to localized PCa. In addition, *BRD7* loss is likely part of a larger heterozygous deletion event centered around the *CYLD* gene. We previously demonstrated that *CYLD* is frequently deleted in CRPC-NE (Beltran et al., PMID 26855148 and 32091413) and it is located close to *BRD7*. Overall, we believe that the above genomic alterations in *BRD7*, *PBRM1* and *SMARCD1* are unlikely to carry a functional significance.

Changes to manuscript #6. We have included the above results in the Results section of the manuscript and as **Supplementary Fig. S1.2.**

7) Given the mutual exclusivity between *BRG1* and *BRM*, it is odd to see *BRG1* pulled down *BRM* in Fig. S1.6. Are they in the same complex or this is due to high background in IP analysis?

Response #7. We fully agree with the Reviewer. We interpret this result as background in the IP analysis and could be a non-specific finding.

Changes to manuscript #7. We have excluded Fig. S1.6 from the manuscript.

8) In Fig. S1.7, does *BRG1* knockdown in CRPC-NE cells affect neuronal genes?

Response #8. Fig. S1.7 refers to the effects of REST knock-down on neuronal genes. The suggestion to also investigate the effects of *SMARCA4* knock-down on neuronal genes is interesting. However, as mentioned in the discussion section of our manuscript, we would like to emphasize that “neuroendocrine genes” is a rather wide concept which has to be subdivided into the expression of terminal neuronal genes (e.g. *SYP*, *CHGA*) in CRPC-NE, and other characteristics of CRPC-NE including dedifferentiation, AR signaling indifference, acquiring “stem cell”-like features (e.g. *SOX2* expression), and high proliferation

(Davies et al., PMID 29460922). There is increasing evidence that de-repression of “terminal” neuronal genes in PCa cells is not sufficient to model other critical steps of neuroendocrine lineage plasticity in CRPC-NE (Labrecque et al., PMID 31361600). Likewise, an aggressive, double-negative subtype of CRPC has recently been identified, which is characterized by the lack of expression of both AR and of terminal neuronal markers (Bluemn et al., PMID 29017058). **As such, we believe that high SMARCA4 expression in CRPC-NE is more related to stem cell-like features and/or to proliferation, than to the expression of terminal neuronal markers.** This is in keeping with our Supplementary Fig. S1.6 (also shown below), where we report that clusters of CRPC-NE organoid cells with high expression of SOX2, a neural stem cell marker, show higher expression of SMARCA4 and SMARCC1, and lower expression of terminal neural markers (e.g. SYP). To further support this hypothesis, we performed single cell RNA-seq data from two CRPC-NE organoids (MSKCC PCa1 and 16). We observed a significant correlation between SMARCA4 and SOX2 expression at a single cell level (shown below).

We also analyzed bulk RNA-seq data from 16 MSKCC PCa organoids and 2 WCMC PCa organoids. Pearson correlation analysis of these data, summarized in the scatter plot below, showed that SMARCA4 expression showed a tendency towards a positive correlation with SOX2 (a neural stem cell marker) and with SYP (a terminal neuronal marker). In line with this, there was a trend towards a negative correlation between SMARCA2 and SOX2, and between SMARCA2 and SYP. This analysis further revealed that although MSKPCa1 and MSKPCa16 organoids (which were used for the single cell RNA-seq experiment) are classified as CRPC-NE based on their transcriptomic NEPC score, they show very high expression of SOX2, but very low expression of SYP. Thus, in these two models, high SMARCA4 expression seems to be correlated with the expression of SOX2, rather than with terminal neuronal genes such as SYP. These findings support our hypothesis that high SMARCA4 expression in CRPC-NE is rather related to stem cell-like features and to proliferation, and not only to the expression of terminal neuronal markers.

Supplementary Fig. S1.6. Immunohistochemistry in a CRPC-NE organoid (3D culture) after FFPE processing. SMARCA4 (BRG1) expression is particularly high in clusters that also show high SOX2 expression (arrow), while also showing negativity for the terminal neuronal differentiation marker synaptophysin (SYP). SMARCA2 (BRM), BAF53B, and SYP are expressed in another population of cells.

New Supplementary Fig. S1.9. Single cell RNA-seq data from two CRPC-NE organoids in 3D growth, demonstrating significantly higher SMARCA4 expression levels in cells that also show high SOX2 expression (defined as superior to mean SOX2 expression in each experiment); Wilcoxon test.

New Supplementary Fig. S1.10. Pearson correlation analysis of gene expression values (FPKM) from bulk RNA-seq in patient-derived PCa organoids, presented as a scatterplot. Each organoid is color-coded according to its transcriptomic NEPC score, whereby a score >0.4 indicates a CRPC-NE phenotype and a score <0.4 indicates a CRPC-Adeno phenotype (Beltran et al., PMID 26855148). The two organoids used in the single cell RNA-seq experiment (Supplementary Fig. S1.9.) are indicated in red font.

Changes to manuscript #8. These results have been added to the manuscript (Results, **Supplementary Figs. S1.9 and S1.10**). Some of the above points have been added to the Discussion section of the manuscript.

9) Given the role of REST in regulating BAF53B and neuronal genes, it is informative to examine whether BRG1 knockdown affects REST expression.

Response #9. We thank the Reviewer for this great suggestion. We analyzed the RNA-seq data generated in LNCaP and 22Rv1 cells with BRG1 knock-down. We did not observe any significant changes in REST transcript levels upon BRG1 knock-down compared to the Scrambled control in LNCaP cells (shown in figure below)

New Supplementary Fig. S2.4. Gene expression levels of the REST gene upon SMARCA4 knock-down, assessed by RNA-seq in LNCaP and 22Rv1 cells. There is no statistically significant difference in REST expression between the SMARCA4 knock-down condition and the control (Scrambled siRNA). Wilcoxon test.

Changes to manuscript #9. We have added this information to the Results section and to Supplementary figures (**Supplementary Fig. S2.4**).

10) Expression status of BRG1 and BRM should be included in Fig. 1e and BRM expression should be included in Fig. 4d.

Response #10. We fully agree with the Reviewer that completeness is important.

The BRG1 and BRM protein levels across cell lines were queried in this experiment (shown in the figure below). The data were not included in Fig. 1e, since we wanted it to focus on BAF53B and BAF53A subunit expression. Unlike in patient samples, we did not observe a clear trend of high BRG1/low BRM expression in cell lines (figure below). The authors believe that this may be due to a tightly linked relationship of the SWI/SNF complex and proliferation, and as such, BRG1 and BRM levels across cell lines may be partly affected by the growth rate and culture conditions (e.g. media composition) for each cell line.

Reviewer #2 (Remarks to the Author)

This study by Rubin and coworkers investigated the role of the mammalian SWI/SNF chromatin-remodeling complex in neuroendocrine prostate cancer. This is achieved through integrating a large collection of clinical prostate cancer datasets with some functional studies in organoids and cell lines. The results show that mammalian SWI/SNF subunits are deregulated and indicate tumor-promoting roles in neuroendocrine prostate cancer. Consistent with several previous studies, high SMARCA4 expression was observed to be associated with aggressive types of prostate cancers. Finally, the authors report that the SWI/SNF complexes interact with different lineage-specific factors in neuroendocrine prostate cancer compared to prostate adenocarcinoma.

Overall, this study is well carried out and is very complete in terms of clinical correlation analysis of the SWI/SNF complex performed. The following suggestions are for improvement.

Response: Thank you for these supportive comments.

1. In Figure 1b, compared to CRPC-Adeno, a significant increase of LOH fraction in CRPC-NE was observed for three genes: BRD7, SMARCD1, and PBRM1. The authors may provide gene expression data to show whether LOH events correlate with their transcript levels.

Response #1. This question resembles question #6 asked by Reviewer 1. Please kindly refer to the answer above.

2a. The results in Figure 1 clearly showed that the SMARCA4 was significantly upregulated while SMARCA2 was downregulated at both transcriptional and protein levels in CRPC-NE. For characterizing the effects of SMARCA4 and SMARCA2, why the authors did not use CRPC-NE organoid model as that of the other SWI/SNF subunits BAF53B and BAF45B? NE-like PC3 cell line (PMID: 21432867) may be a good model for this purpose.

Response #2a. This is an excellent point and we recognize that these data would have been of great interest in this study. Indeed, we performed additional transient SMARCA4 and SMARCA2 knock-down experiments in the CRPC-NE WCM155 cell line, as well as SMARCC1 and SMARCC2 knock-down 22Rv1 cells, to assess the effects of knock-down on cell growth. We present these data in our response to question #4 asked by Reviewer 1. Please kindly refer to the answer above. The 22Rv1 cell line has been

shown to display some features of the CRPC-NE phenotype (Bluemn et al., PMID 29017058). As such, the RNA-seq data we generated in 22Rv1 cells upon SMARCA4 and SMARCA2 knock-down could be of interest, even if they do not replace a similar experiment in a CRPC-NE cell line with a more pronounced phenotype.

Changes to manuscript #2. To allow for better granularity in the interpretation of the presented data, we have pointed out the presence of some CRPC-NE features in 22Rv1 cells in the manuscript. Results of SMARCC1 and SMARCC2 knock-down 22Rv1 cells have been added to the manuscript.

2b. Meanwhile, it would be interesting to know to what degree the impact of SMARCA4 knock-down while overexpressing SMARCA2 in CRPC-NE cell lines, and organoids if possible. To solidify the findings, in addition to siRNA-mediated knock-down for SMARCA4 and SMARCA2, the complementary assays using shRNA and/or CRISPR-Cas9 are encouraged.

Response #2b. Regarding experiments with SMARCA2 overexpression, this question is similar to question #1 asked by Reviewer 1. Please kindly refer to the answer above. Regarding complementary assays using shRNA and CRISPR-Cas9, we fully agree that this experiment would strengthen our findings. Unfortunately, in the current context, we are unable to perform these additional experiments on organoid models in a timely manner.

Changes to manuscript #2b. No changes were made to the manuscript regarding this point.

3. What is also missing is to analyze transcriptional and epigenetic reprogramming in CRPC-NE versus prostate adenocarcinoma cells while depleting the core SWI/SNF subunit BRG1 or SMARCC1. In parallel RNA-seq and chromatin profiling such as ATAC-seq may be useful to address lineage-specific roles of the SWI/SNF complex corroborating with their lineage-specific interaction partners in particular transcription factors

Response #3. These are excellent suggestions. However, given that the goal of this study was to provide a general overview of SWI/SNF deregulation in the context of CRPC-NE, studying epigenetic changes upon manipulation of SWI/SNF subunits may be beyond the scope of this manuscript. Nevertheless, we plan to pursue this issue for future publications in follow-up studies.

Changes to manuscript #3. No changes were made to the manuscript regarding this point.

Minor comments:

1. Minor errors, such as 'previously described30' in line 4 page 21, 'Hochberg method99' in line 4 page 27; 'and in CRPC-Adeno versus CRPC-Adeno (c)' in Supplementary Figure S1.1, two 'Lowest' in Supplementary Figure S3.2b.

Response: Thank you very much for pointing this out. These errors have been corrected.

2. It is likely to miss proper references in line 1 page 10 "significant decrease in PCa cell growth, in line with previous studies".

Response: Thank you very much for pointing this out. This has been rectified.

REVIEWER COMMENTS

Reviewer #1 (Remarks to the Author):

This reviewer appreciates the efforts by the authors in trying to address the concerns raised in the initial review during these difficult times. The revised version of the manuscript is improved with additional experiments, database mining and clarifications. However, one would like to see the following straight forward experiments performed to support the functional significance and potential mechanistic link as proposed:

1. To examine ARSi treatment responses in both CRPC-NE and non-CRPC-NE cell lines with BRG1 knockdown. This is critical to support the biological significance of the study and the database mining results as presented are not informative.
2. Without ChIP-seq data (given the current difficult time), one would like to see a validation of interaction between lineage-specific factors and SWI/SNF complex with comparison between CRPC-NE (NKX2.1, MAP2, VGF, MTA1 and CHD4) and non-CRPC-NE by immunoprecipitation and immunoblotting.

Minor:

1. Supplementary Figure 1.7 title label on the figure should be "ACTL6A (BAF53A).
2. Supplementary Figure 2.6, siSMARCC2 results should be described.

Reviewer #2 (Remarks to the Author):

The authors have addressed all the major questions raised by both referees and extensively revised the manuscript within just two months. The revised manuscript includes additional experiments and data analysis. This referee agree the SWI/SNF dependent transcriptional and epigenetic reprogramming between CRPC-NE versus prostate adenocarcinoma cells can be addressed in the future investigation, and thus recommend the manuscript is acceptable for Nature Communications.

NCOMMS-20-05343A: Point by Point Response

Reviewers' comments:

Reviewer #1 (Remarks to the Author):

This reviewer appreciates the efforts by the authors in trying to address the concerns raised in the initial review during these difficult times. The revised version of the manuscript is improved with additional experiments, database mining and clarifications. However, one would like to see the following straight forward experiments performed to support the functional significance and potential mechanistic link as proposed:

Response: We thank the Reviewer for these thoughtful and supportive comments. Indeed, these have been challenging times to coordinate laboratory experiments.

1. To examine ARSi treatment responses in both CRPC-NE and non-CRPC-NE cell lines with BRG1 knockdown. This is critical to support the biological significance of the study and the database mining results as presented are not informative.

Response #1. We appreciate this comment and agree that this experiment could be informative. To address this, we performed transient BRG1 knock-down in the CRPC-NE cell line WCM154 (developed from a patient derived organoid and described in Puca et al., PMID 29921838), the prostatic adenocarcinoma cell line LNCaP, and the LNCaP-derived CRPC-Adeno cell line C4-2, which shows only minimal sensitivity to ARSi. We treated these cells with 15uM of the ARSi agent enzalutamide or DMSO to observe effects on cell proliferation. Cells were grown for 156h under these conditions in a Live-Cell imager. Knock-down efficiency was confirmed by immunoblotting after 96h. The results are presented for the Reviewer below. In the CRPC-NE cell line WCM154, BRG1 depletion did not restore sensitivity to enzalutamide treatment. LNCaP cells were exquisitely sensitive to enzalutamide treatment, as expected, both upon transfection with non-targeting siRNA and upon BRG1 depletion. Lastly, in the CRPC-Adeno cell line C4-2, combined depletion of BRG1 and ARSi treatment had the most potent negative effect on cell growth.

Figure: (Top) Immunoblot showing siRNA-mediated knock-down efficiency of SMARCA4 with DMSO (0.1%) or with 15uM Enzalutamide in three different PCa cell lines LNCaP, C4-2 and WCM154 after 96h hours, Actin serves as loading control. The immunoblot displays one replicate of a total of three, done for each individual experiment. (Bottom) Growth curves showing confluence upon siRNA-mediated SMARCA4 knock-down combined with DMSO or 15uM Enzalutamide treatment until 156 hours, in three different PCa cell lines. Three pooled independent experiments are shown. Error bars, SEM.

Changes to manuscript #1. Based on this negative result, we favor not including this in the current manuscript. We have noted the results in the result section and added (data not shown). We are happy to include this in the supplement if this is preferred.

2. Without ChIP-seq data (given the current difficult time), one would like to see a validation of interaction between lineage-specific factors and SWI/SNF complex with comparison between CRPC-NE (NKX2.1, MAP2, VGF, MTA1 and CHD4) and non-CRPC-NE by immunoprecipitation and immunoblotting.

Response #2. Analogous to the Co-IP followed by immunoblotting previously performed for HOXB13 in the CRPC-Adeno cell line LNCaP-AR, we performed the same experiment in the CRPC-NE cell line NCI-H660 and in LNCaP-AR cells to visualize the interaction between SMARCC1 and the factors NKX2.1, MAP2, VGF, MTA1 and CHD4. In NCI-H660 (NEPC model), factors NKX2.1, VGF, MTA1 and CHD4 were present by immunoblotting in the SMARCC1 Co-IP condition, but were either absent or only detected at very low levels in the IgG condition (control). The MAP2 signal was very weak and thus, it was not possible to draw any conclusions from this immunoblot; we therefore prefer not to include this inconclusive result for MAP2 in the manuscript. In contrast, in LNCaP-AR cells (CRPC-Adeno model), these factors were not detectable in the SMARCC1 Co-IP condition, even though some of them were present in the nuclear fraction (input). These findings are in line with our mass spectrometry results (Figure 4), and support the hypothesis that these interactions only manifest in the CRPC-NE phenotype, but not in the CRPC-Adeno phenotype. We thank the Reviewer for suggesting this additional validation experiment.

New Supplemental Fig. 4.2. Co-immunoprecipitation experiment of BAF155 (SMARCC1) in the CRPC-Adeno cell line LNCaP-AR and in the CRPC-NE cell line NCI-H660, with immunoblotting for SMARCA4, MTA1, MAP2, CHD4, VGF, NKX2.1 (TTF1) and H3. IP = immunoprecipitation; SN = supernatant.

Changes to manuscript #2. These results have been included in the Supplementary Material as Supplemental Fig. 4.2 (except for the MAP2 band). Given that some of the proteins (e.g. VGF) were detected in the nuclear fraction of LNCaP-AR cells with the new detection method used in this experiment (HRP-based), we removed the previous Figure 4.d from the manuscript to avoid discordance, and because we believe these results are more sensitive than the previously performed immunoblotting from Fig.4d.

Minor:

1. *Supplementary Figure 1.7 title label on the figure should be "ACTL6A (BAF53A).*

Response: Thank you for pointing out this error. We corrected it in the supplemental of the manuscript.

2. *Supplementary Figure 2.6, siSMARCC2 results should be described.*

Response: Thank you for this comment. We included this information in the manuscript.

Reviewer #2 (Remarks to the Author)

The authors have addressed all the major questions raised by both referees and extensively revised the manuscript within just two months. The revised manuscript includes additional experiments and data analysis. This referee agree the SWI/SNF dependent transcriptional and epigenetic reprogramming between CRPC-NE versus prostate adenocarcinoma cells can be addressed in the future investigation, and thus recommend the manuscript is acceptable for Nature Communications.

Response: We thank the Reviewer for the supportive comment.

REVIEWERS' COMMENTS

Reviewer #1 (Remarks to the Author):

The authors have addressed the remaining comments and modified the conclusions accordingly. In particular, SMARCA4 knockdown does not affect response to ARSi in CRPC-NE cell line WCM154. The manuscript is now acceptable for publication.